# Single-neuron correlates of visual consciousness in human lateral occipital complex

Michaël Vanhoyland [1,2,3] ✉, Peter Janssen [2] & Tom Theys [1,3]

Conscious perception, a critical aspect of human cognition, is assumed to emerge from a complex network of interacting brain regions that transmit information via feedforward and recurrent pathways. This study presents single- and multiunit recordings from the human lateral occipital complex (LO), a key region for shape and object recognition, during three distinct perceptual paradigms: backward masking, flash suppression and binocular rivalry. Stimulus awareness increased decoding accuracy and decoders assigned higher probabilities to the consciously perceived stimulus during periods of dichoptic stimulus presentation. These findings highlight the intricate neural mechanisms underlying visual awareness and show that LO responses predominantly align with subjective phenomenology, offering new insights into the neural correlates of visual consciousness.

Between the instant the world is projected onto our eyes and the moment we become conscious of that world, a complex network of interacting brain regions comes into play, passing information to one another in feedforward and recurrent ways[1,2]. Not all sensory input reaches the final state of conscious perception. At the single-neuron level, the neural correlates of conscious visual perception in humans remain elusive.

Specific paradigms have been developed to dissociate stimulus presentation from conscious perception. These include backward masking[3], where the perception of a briefly presented stimulus is suppressed from conscious perception by the rapid presentation of a masking image; flash suppression[4], where perception of an image is suppressed by the sudden presentation of a mask to one eye while the target image remains present in the other eye; and binocular rivalry[5,6], where each eye is presented with a different stationary image, leading to alternating periods of perceptual dominance. These paradigms differ in the way they alter perception: backward masking and flash suppression involve external experimental manipulations, whereas binocular rivalry results from internal perceptual conflict[5].

Non-human primates have been used to investigate visual awareness at the single-neuron level across brain regions[4–8]. In humans, however, single-unit recordings are mostly restricted to areas accessible via depth electrodes, typically involving deep and medial structures such as the medial temporal lobe (MTL) and medial frontal lobe[3,9]. Single-unit recordings from visually responsive regions on the brain's convexity, including lateral occipital cortex, are virtually non-existent due to the inherent properties of most clinical microelectrodes, which are typically attached to depth electrodes. Previous studies in non-human primates have shown that the proportion of neurons following the perceptual report increases along the visual processing hierarchy, with approximately 20% of neurons in primary visual cortex and up to 60–90% in the inferotemporal cortex (ITC) showing activity modulated according to the percept[10,11]. How this translates to the human visual system remains unresolved.

We recorded single- and multiunit neural activity from five microelectrode arrays implanted in lateral occipital cortex (LO) of four neurosurgical patients. Using linear decoders, we demonstrated that a population of LO neurons tracks perception during backward masking, binocular rivalry, and flash suppression. By employing multiple paradigms, we could disentangle various aspects of the relationship between conscious perception and neuronal activity. During flash suppression, both individual neurons and population responses predominantly encoded the perceived stimulus, but a subset of neurons did not modulate activity according to the percept. In backward

[1]Research group Experimental Neurosurgery and Neuroanatomy, KU Leuven, Leuven, Belgium. [2]Laboratory for Neuro- and Psychophysiology, Department of Neurosciences, KU Leuven and the Leuven Brain Institute, Leuven, Belgium. [3]Department of Neurosurgery, UZ Leuven, Leuven, Belgium. ✉ e-mail: michael.vanhoyland@kuleuven.be

masking, decoding accuracy improved markedly when the target stimulus was consciously perceived. In binocular rivalry, neuronal activity reflected the perceived image, and decoders started to signal the perceived stimulus approximately 1.5 s before self-reported perceptual changes. These findings could point towards a critical role for LO in the neural processes underlying conscious visual perception, bridging the gap between sensory input and subjective awareness.

## Results

### Visual selectivity

The lateral occipital complex is functionally heterogeneous[12–14]. To characterize the functional properties of individual recording sites, we recorded 258 visually responsive multiunits from 5 arrays (51 in A1, 92 in A2, 56 in A3, 35 in A4, and 24 in A5) while patients watched scrambled and non-scrambled images of people and animals. All arrays responded stronger to the non-scrambled images, as visualized in Fig. 1B (A1: $p = 6.46 \times 10^{-7}$, A2: $p = 1.37 \times 10^{-81}$, A3: $p = 2.61 \times 10^{-310}$, A4: $p = 1.28 \times 10^{-106}$, A5: $p = 1.49 \times 10^{-86}$; one-sided independent samples t-test comparing z-normalized net spike rate within the response window). Discriminability between non-scrambled and scrambled images was quantified using the d-prime index (d′), where a value of 0 indicates no discriminability. A1 showed minimal discriminability (d′ = 0.26), A2 moderate (d′ = 1.08), and A3, A4, and A5 strong discriminability (A3: 2.70, A4: 2.53, A5: 1.79; Figs. 1A, C and S2 and Table S1). Population response latencies were notably rapid in A1 and A2, initiating at 65 ms and 85 ms, respectively (Fig. S3). In contrast, A3 and A5 exhibited slower responses, starting at 135 ms, whereas A4 demonstrated a markedly delayed response at 255 ms. The weak selectivity observed in A1 was limited to the latter part of the response

(230–370 ms), whereas selectivity emerged at 135 ms for A2 and A3, at 145 ms for A5, and at 255 ms for A4.

To assess the presence of category selectivity, we recorded 314 visually responsive multi-units (52 in A1, 92 in A2, 72 in A3, 66 in A4, and 32 in A5) while presenting 10 different object categories. Arrays A1 to A3 exhibited a preference for bodies or faces in a small number of multiunits (A1: 10%, A2: 16%, A3: 18%; one-way ANOVA, $p < 0.05$), whereas a significant proportion of units in A5 (41%) and A4 (85%) were category-selective, with a strong preference for bodies in A4 (77% of responsive units were body-selective, Tukey's HSD, $p < 0.05$) (Figs. 1D, S4, and S5 and Table S1). A linear decoder could reliably classify scrambled from non-scrambled images and differentiate object categories across all arrays except A1 (Figs. 1E and S6).

Based on response latency and selectivity profiles, A1 appears to be in an inferoposterior subregion of LO with weak shape selectivity, at least for the stimulus sets used here, as visualized in Fig. 1A. In contrast, A2–A5 lie within shape-selective regions of LO[12,15]. Furthermore, the response profile of A4 closely matches that of the extrastriate body area (EBA), as shown in a previous study[16,17].

### Backward masking

During backward masking, target stimuli—categorical images of faces, bodies, objects, or naturalistic images—were rendered invisible by presenting a masking image (Mondrian) in rapid succession (see Fig. 2 for paradigms). The strength of perceptual suppression depends on the delay between target and mask. Neuronal responses under these conditions are shaped by both the target and the masking image. To isolate their respective contributions, we recorded responses to the different stimulus categories (presented for

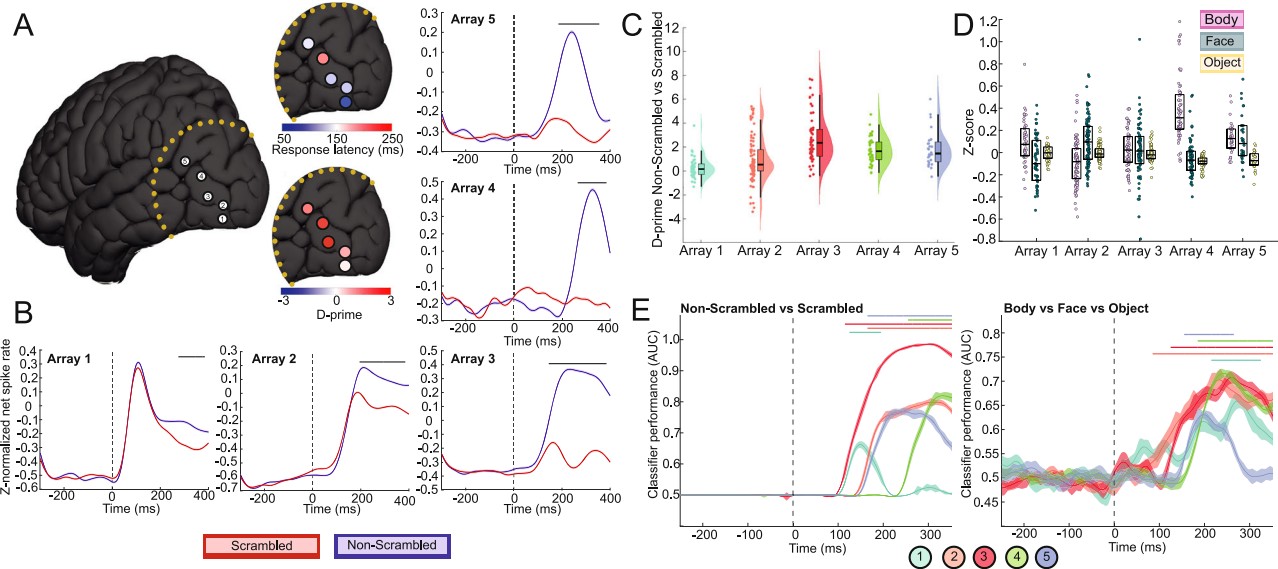

**Fig. 1 | Multielectrode array recording sites and stimulus selectivity for each array. A** Anatomical locations of Utah arrays projected on a common MNI template of the brain. Adapted from Bougou, V., Vanhoyland, M., Bertrand, A. et al. Neuronal tuning and population representations of shape and category in human visual cortex. Nat Commun 15, 4608 (2024)[15]. MNI coordinates (X, Y, Z) are as follows: A1 (42, −76, −10), A2 (−35, −89, −8), A3 (−41, −83, 9), A4 (−38, −84, −5), and A5 (51, −66,19). The average response latency and d-prime comparing non-scrambled and scrambled images per array are presented separately. Arrays plotted on cortical renderings of individual subjects are provided in Supplementary Fig. S1. **B** Average z-normalized net MUA responses to non-scrambled and scrambled images for all visually responsive channels. Horizontal bars indicate significance (non-scrambled > scrambled, $p < 0.001$, one-sided permutation test). Dotted vertical lines mark stimulus onset. Shading represents standard error (N = number of trials per condition x number of responsive MUA). **C** Combined box, violin, and scatter plot showing the d-prime values (non-

scrambled vs scrambled) within the response window for all responsive multi-units per array. Box plots indicate median (middle line), 25th, 75th percentile (box), and box limits ± 1.5 × interquartile range (whiskers). **D** Combined box and scatter plot showing the z-normalized net responses within the response window to the human body, human face, and object categories for all visually responsive multiunits per array. Box plots indicate median (middle line), 25th, 75th percentile (box). Strong body preference is evident for array 4. **E** Classifier performance over time for scrambled versus non-scrambled images (AUC) **(left)** and for multiclass classification among human body, human face, and objects (macro-averaged AUC) **(right)**. Lines and shading represent the mean ± standard error classification AUC from 10 decoding repetitions. Colored horizontal bars indicate above-chance classification compared to random label shuffling ($p < 0.001$, one-sided permutation test). Vertical dotted lines mark stimulus onset. Times on the x-axis represent the middle of the 100 ms intervals used for classification. Numbers refer to array number.

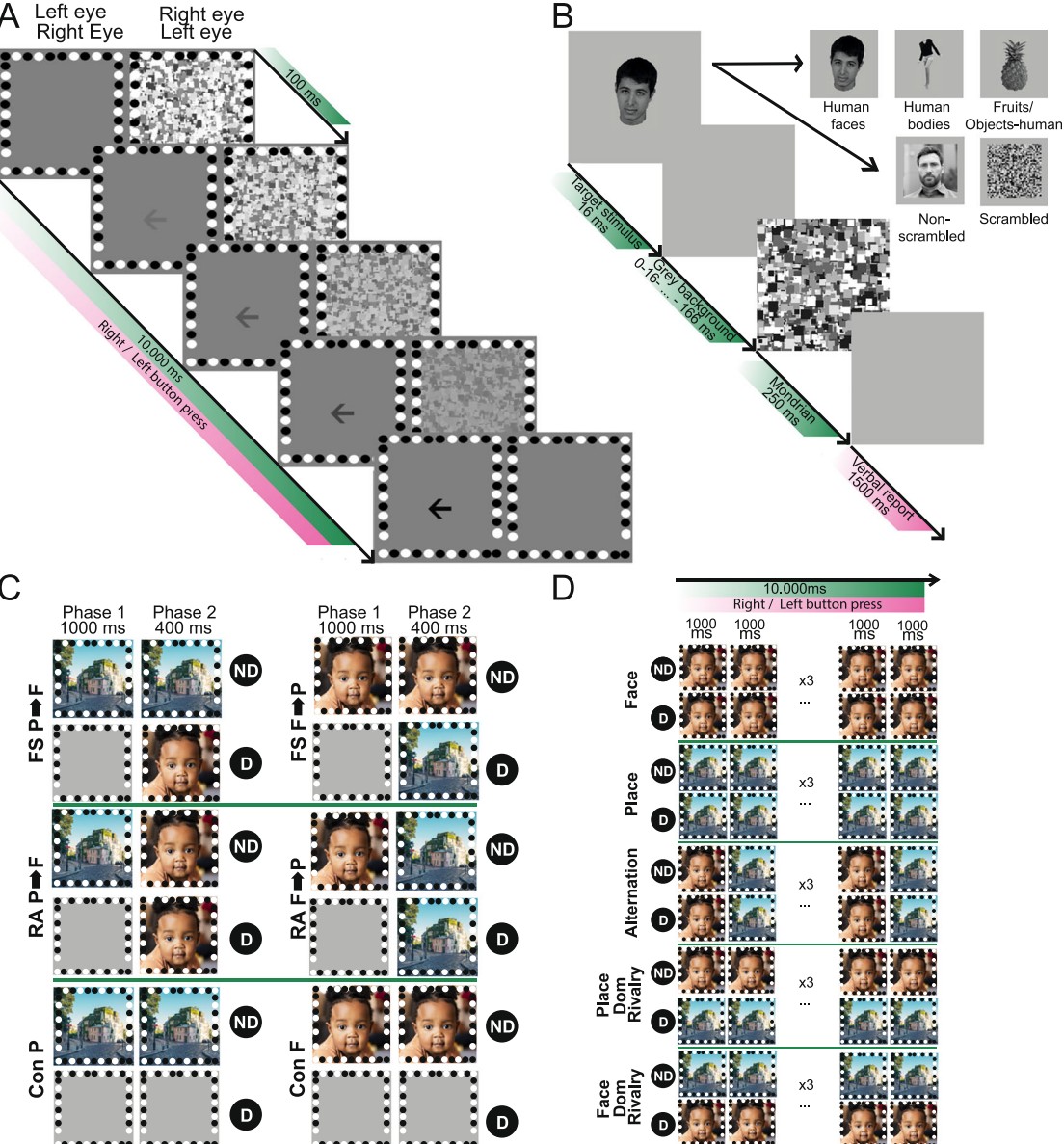

**Fig. 2 | Experimental paradigms. A Eye dominance testing via breakage of Continuous Flash Suppression (b-CFS). A** left- or right-pointing arrow is presented in either the left or right eye, positioned above, at, or below the fixation marker. The contrast of the arrows linearly increases over 10 s, while the Mondrians' contrast linearly decreases. Mondrians alternate at 10 Hz. **B Backward masking.** Target stimuli (scrambled and non-scrambled images, or categorical images) were shown for 16 ms, followed by a masking Mondrian for 250 ms. A gray background is displayed between the target stimulus and the Mondrian for a variable duration to adjust the suppressive strength. After the Mondrian, patients had 1500 ms to verbally report their percept. Categorical stimuli were published in Popivanov, I. D., Jastorff, J., Vanduffel, W. & Vogels, R. Stimulus representations in body-selective regions of the macaque cortex assessed with event-related fMRI. Neuroimage 63, 723–741 (2012), Copyright Elsevier[73]. The non-scrambled image is a stock photo (iStock.com/Liubomyr Vorona), comparable to the images that were used from the IAPS database[71]. **C Flash suppression.** Two naturalistic images, a preferred (face) and nonpreferred (place) image, were used. Perception is shifted from one image to the other by flash suppression (FS F→P and FS P→F), real alternation (RA F→P and RA P→F), or no perceptual shift (Con). The first 1000 ms are defined as phase 1, and the last 400 ms as phase 2. **D Binocular rivalry.** Same stimuli as in flash suppression. The Rivalry condition was performed with the preferred stimulus in the dominant (Face Dom Rivalry) and the nondominant eye (Place Dom Rivalry). Both stimuli were also presented binocularly for the entire duration of the experiment or alternated binocularly at a frequency of 1 Hz. Each trial lasted 10 s for patients 2, 3, and 4, and 24 s for patient 1. "D" and "ND" represent Dominant and Non-Dominant eye in (**C** and **D**). The face and place image in (**C** and **D**) are stock photos (face: iStock.com/Djavan Rodriguez; place: iStock.com/Iakov Kalinin).

16 ms) during passive fixation as well as to isolated Mondrians (presented for 250 ms) in arrays A2, A3, A4, and A5, and trained linear decoders to predict the stimulus category per trial. Classification of the briefly presented, unmasked stimuli was only above chance in A2 and A4 (Fig. S7; $p < 0.01$; one-sided permutation test), suggesting that significant decoding of perceived targets during backward masking could only be expected in these two arrays. Responsiveness to the masking Mondrians varied across arrays, with 99% of the MUAs responding to the masking Mondrian in A2,

compared to 42%, 5% and 25% for A3, A4, and A5, respectively (Table S2).

Next, we investigated the interactions between target and mask across varying delays, identifying two distinct response patterns (Figs. S8–10). In arrays with strong responsiveness to the Mondrians (A1 and A2), population activity exhibited two distinct peaks corresponding to target and mask separately, but only at longer delays (≥132 and ≥83 ms in A1 and A2, respectively) (Silverman's test, $p < 0.05$, Table S6). At shorter delays, these peaks gradually merged. In contrast,

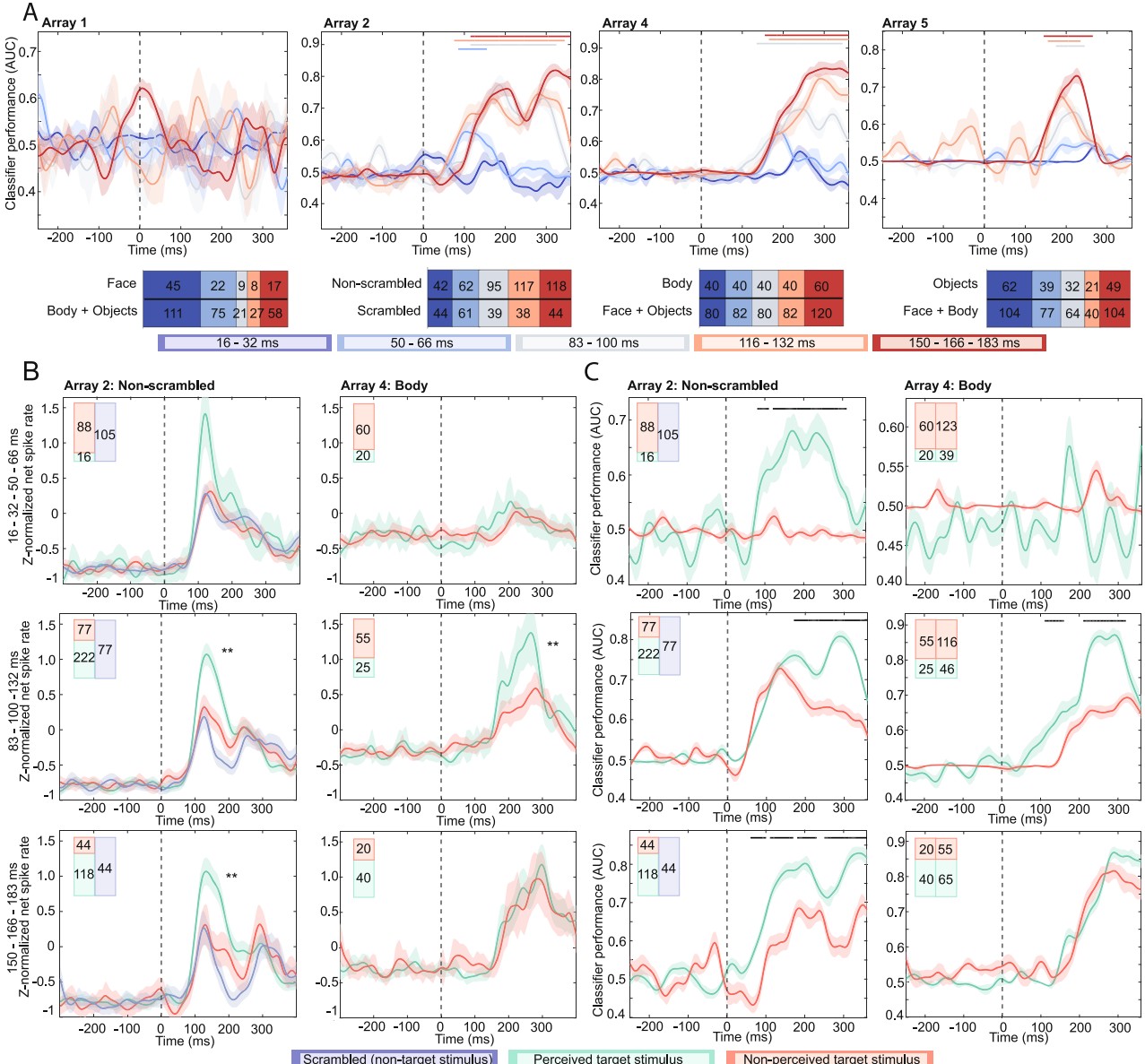

**Fig. 3 | Backward masking. A** Linear decoder results of the preferred class for different delays, regardless of stimulus perception. Time on x-axis represents middle of 100 ms bins used for decoding. Lines and shading represent the mean ± standard error classification AUC from 10 decoding repetitions. Horizontal bars represent significant decoding performance compared to random label shuffling ($p < 0.001$, permutation test with at least five consecutive significant time points). The bars below the plots show the number of trials per condition. **B** Average z-normalized spike rate from all channels selective for the preferred class (preferred class > all other classes; $p < 0.01$, one-sided permutation test). Shading signifies standard error. Asterisks indicate significance between perceived and nonperceived target stimuli within the response window (A2 [130–280], A4 [200–350]) (*$p < 0.05$, **$p < 0.01$, one-sided independent samples $t$-test). **C** Decoding results for preferred class for perceived and nonperceived trials. Lines and shading represent the mean ± standard error classification AUC from 10 decoding repetitions. Time on x-axis represents middle of 100 ms bins used for decoding. Horizontal bars represent significantly better decoding performance for perceived stimuli ($p < 0.01$, one-sided independent samples $t$-test). The same delays are grouped as in (**B**). The bars in the left upper corner of each plot in (**B** and **C**) show the number of perceived (green) and nonperceived (red) trials for the preferred class (left) and all other classes (right). For array 2, the other class is scrambled images (blue).

arrays with weak Mondrian responsiveness (A4 and A5) exhibited a single peak, emerging only at longer delays (≥66 ms).

We trained a linear decoder to classify the preferred class under masked conditions (Fig. 3A). To enhance statistical power, trials with similar delays were pooled (see "Methods"). No reliable decoding was possible in A1 (and A3 due to poor signal quality on the day of recording). Classification performance improved with longer delays for all other arrays regardless of whether the stimulus was perceived, starting at 50–66 ms in A2 and A4, and at 83–100 ms in A5 (Figs. 3A and S11A, cross-temporal generalization of the decoding is

visualized in Fig. S12). No reliable decoding was possible for the shortest delays (16–32 ms) in any array.

To further assess the impact of stimulus perception on decoding, we divided the delays into three equal-sized groups (≤ 66 ms, 83–132 ms, and ≥ 150 ms) to mitigate bias from increasing proportions of correct trials at longer delays (Fig. S13). We calculated the average z-normalized responses for all selective single-units (see "Methods") to their preferred class within each delay group ($N_{A2} = 8$, $N_{A4} = 6$ selective units) and decoded the preferred class for perceived and non-perceived trials. Both decoding performance and spike rate were

significantly higher for correctly perceived trials in A2 across all delays, except spike rate ≤ 66 ms (spike rate analysis: ≤ 66 ms: $p = 0.186$; 83–132 ms: $p = 1.53 \times 10^{-5}$; ≥ 150 ms: $p = 3.94 \times 10^{-4}$; one-sided independent samples $t$-test comparing spike rate within the response window; decoding analysis: ≤ 66 ms: $p = 3.12 \times 10^{-4}$; 83–132 ms: $p = 0.007$; ≥150 ms: $p = 3.96 \times 10^{-5}$; one-sided independent samples $t$-test comparing decoding AUC within the response window over 20 repetitions for perceived versus unperceived trials). In A4, spike rates and decoding were only significantly higher in the 83–132 ms delay (spike rate analysis: ≤ 66 ms: $p = 0.824$; 83–132 ms: $p = 2.66 \times 10^{-4}$; ≥ 150 ms: $p = 0.556$; decoding analysis: ≤ 66 ms: $p = 0.870$; 83–132 ms: $p = 2.41 \times 10^{-10}$; ≥150 ms: $p = 0.703$; Figs. 3B, C and S11B and Table S3).

Next, we quantified the percent difference in neural responses between perceived and nonperceived stimuli using the signal loss metric (see "Methods"). Among all responsive units, perceived stimuli evoked 27% and 15% stronger responses in A2 with categorical ($p = 2.01 \times 10^{-6}$) and scrambled vs non-scrambled images ($p = 5.62 \times 10^{-13}$), respectively; 45% stronger responses in A4 ($p = 3.52 \times 10^{-11}$); and 18% in A5 ($p = 0.19$). When considering only selective units, responses were 59% greater in A2 for categorical stimuli ($p = 2.71 \times 10^{-11}$), 39% greater in A2 with scrambled and non-scrambled images ($p = 1.24 \times 10^{-32}$), and 42% greater in A4 ($p = 1.15 \times 10^{-8}$; two-sided independent samples $t$-test comparing net spike rate to the preferred class within a 150 ms window following response latency) (Table S4 and Fig. S14). In contrast, nonperceived target images evoked a non-significant 9% stronger response in A1 ($p = 0.35$), likely reflecting strong responsiveness to masking Mondrians in combination with a larger proportion of unperceived stimuli at short delays where the response of target and mask could be summated. Stronger neural responses for perceived targets were observed as early as 95 ms in A2 and 165 ms in A4, faster than the selectivity latency ($p < 0.05$, one-sided permutation test, Table S4).

Finally, we checked if the prestimulus brain-state was predictive for stimulus perception, as this might indicate attentional effects. A decoder was trained on the 300 ms $z$-normalized baseline raw spike rate from all perceived and nonperceived trials to decode whether target stimuli were perceived. An AUC value of 0.50 indicates that prestimulus activity is similar in both perceived and nonperceived trials, whereas an AUC value greater than 0.50 would suggest that the prestimulus state, at least partially, determines whether a stimulus will be perceived. Across all arrays, the AUC was 0.50 (A1: 0.51; A2, A4, and A5: 0.50), indicating no predictive value from baseline activity (Table S5).

## Flash suppression

During flash suppression, perception of a monocularly presented stimulus is suppressed by the sudden presentation of another stimulus to the opposite eye. The suppressive effect is stronger when the masking image is presented to the dominant eye. We therefore started by assessing ocular dominance through a breaking of continuous flash suppression (b-CFS) paradigm and presented the masking image to the dominant eye during flash suppression afterwards. Patient 1 had no dominant eye and did not participate in the flash suppression experiment, whereas patients 2, 3, and 4 were all right eye dominant (Patient 1: $p = 0.86$; patient 2: $p = 0.005$; patient 3: $p = 6.27 \times 10^{-6}$; patient 4: $p = 1.40 \times 10^{-4}$; two-sided independent samples $t$-test, Fig. S17).

During flash suppression, we used naturalistic images of a face and place, as these elicited selective responses across neurons. We recorded 62 visually responsive single-units on four arrays (A2–A5), 37 of which were face-selective and 7 place-selective (Table S7). To be able to investigate if neuronal firing decreases when the perception of a preferred stimulus shifts to a non-preferred stimulus, we identified 18 face-selective sustained or inhibitory responders across all arrays that maintained an elevated firing rate for the whole duration of monocular face presentation, or decreased the firing rate in response to the non-

preferred place image (see "Methods"). An example of a neuron that sustained its response is presented in Fig. 4A. It modulated its firing rate according to the percept, selectively responding to the face during monocular presentation to the nondominant eye. Despite identical visual inputs after flash suppression (after the second vertical dotted line in Fig. 4A)—namely a face and non-face image to opposite eyes—neural activity was significantly different ($p = 2.48 \times 10^{-7}$; two-sided independent samples $t$-test comparing net spike rate within the response window). The neuron fired vigorously only when perception shifted from place to face (bold blue line in Fig. 4A) and ceased firing when perception shifted from face to place (bold red line).

On the population level, the average $z$-normalized response for all face-selective units yielded similar results. Compared with the continuation of the initial image, spike rate decreased when perception shifted from face to place and increased when it changed from place to face, irrespective if this perceptual shift was obtained through real alternation or flash suppression (Figs. 4C, D, S18, and S19, all arrays combined, face to place: $p = 1.29 \times 10^{-24}$, place to face: $p = 2.09 \times 10^{-33}$, two-sided independent samples $t$-test). However, real alternation induced stronger changes in neural activity compared to flash suppression (Fig. S19, all arrays combined: face to place: $p = 7.96 \times 10^{-6}$, place to face: $p = 2.13 \times 10^{-40}$, two-sided independent samples $t$-test). We investigated whether this was due to a smaller number of neurons responding to flash suppression, or due to a weaker response per neuron. Approximately 64% of selective neurons (23 out of 36) followed the percept (see "Methods" for criteria), and all but one neuron (35 out of 36) responded weaker when perception of the preferred image was induced through flash suppression than when real alternation occurred (Fig. S18C–F). Similarly, 69% of sustained or inhibitory responding neurons (11 out of 16) followed the percept, and 94% (15 out of 16) decreased its firing rate less during flash suppression compared to real alternations when perception shifted from the preferred to the nonpreferred stimulus. Again, neural activity during both flash suppression paradigms—opposite perception despite identical visual input—was significantly different (all arrays combined, $p = 1.69 \times 10^{-27}$, two-sided independent samples $t$-test).

We trained linear decoders to study whether the population responses would be classified as the newly introduced stimulus during flash suppression. For this, we trained a decoder on the 100 ms interval that yielded the highest AUC for classifying both images, and then predicted for all intervals over time the probability that a face was presented during each interval. The predictions from the decoder yielded results similar to those of the spike rate analysis. Both images could be clearly distinguished from each other by linear decoders (peak AUC for A2: 98%, A3: 98%, A4: 92%, A5: 78%; Fig. S21). Figure 5A (and for average face probabilities for all conditions per array see Fig. S20) shows that for all arrays except A5, and on a trial-by-trial basis, when perception shifted from place to face through flash-suppression, the decoders classified neuronal activity as more likely to belong to a face percept, despite continued presentation of a place in the nondominant eye (A2: $p = 5.36 \times 10^{-6}$, A3: $p = 1.69 \times 10^{-7}$, A4: $p = 4.95 \times 10^{-4}$, A5: $p = 0.0027$; one-sided independent samples $t$-test comparing FS P→F to Con P within the response window). Similarly, when perception shifted from face to place, the decoder classified neuronal activity as more likely belonging to a place percept in the arrays that sustained a markedly elevated face probability prediction throughout the whole duration of initial face presentation (A2, A3, and A4). (A2: $p = 2.49 \times 10^{-11}$, A3: $6.32 \times 10^{-9}$, A4: $p = 0.0016$; A5: $p = 0.62$, one-sided independent samples $t$-test comparing FS F→P to Con F within the response window). When comparing both flash suppression conditions, the decoder asserted a higher face probability in case the face was perceived (FS P→F) compared to when the place was perceived (FS F→P) (A2: $p = 1.02 \times 10^{-7}$, A3: $p = 6.12 \times 10^{-14}$, A4: $p = 3.06 \times 10^{-8}$, A5: $p = 0.0083$, two-sided independent samples $t$-test comparing FS P→F to FS F→P within the response window).

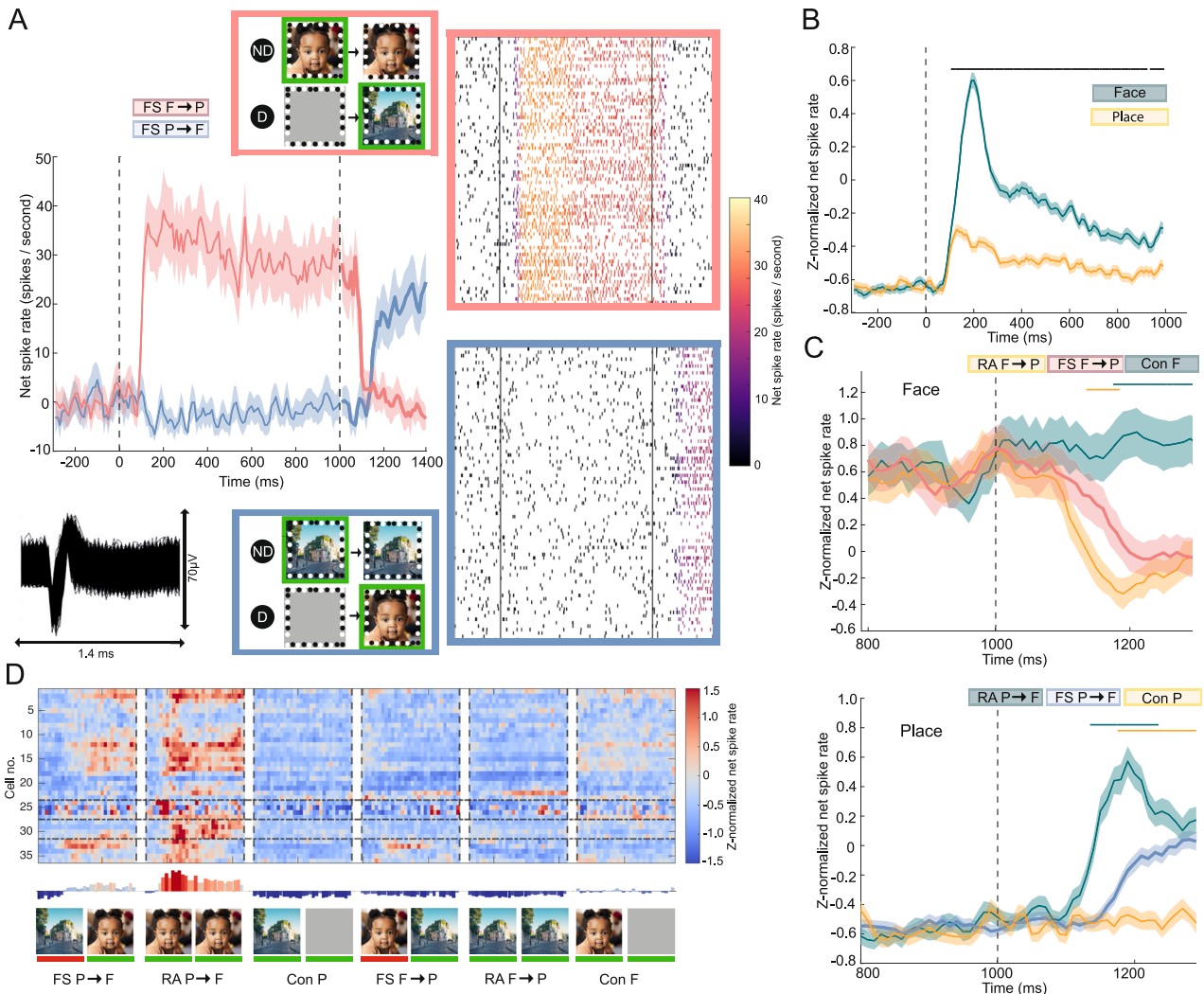

**Fig. 4 | Flash suppression: spike rate analysis. A** Example face-selective sustained/inhibitory responder (array 3). Waveform, raster plot, and average spike rate during flash suppression of initial face perception (red) or initial place perception (blue) are presented. Each row on the raster plots represents a different trial; and vertical lines represent stimulus onset and the moment of flash suppression. Raster plots are color-coded with the average net spike rate. Flash suppression experimental conditions are visualized, with the stimulus that is shown to the dominant (D) and nondominant (ND) eye. Green-framed stimuli are consciously perceived. Bold lines in the line plots represent dichoptic stimulus presentation (same for **C**). **B** Average z-normalized spike rate of all face-selective single-units is significantly different to non-rivalrous face and place stimulus presentation ($p = 5.81 \times 10^{-5}$, one-sided independent samples *t*-test comparing 100–1000 ms after stimulus onset). Horizontal bar indicates significance ($p < 0.001$, one-sided permutation test). Vertical dotted line represents stimulus onset. **C** Average z-normalized net spike rate during phase 2. **Top)** Average response of all face-selective sustained and inhibitory responders during flash suppression (red) and real alternation (yellow) when perception shifts from face to place. Green indicates continued face perception. Spike rate lowers during flash suppression compared to continuation ($p = 1.03 \times 10^{-15}$, two-sided independent samples *t*-test comparing 200–300 ms after stimulus onset), and the decrease is equally strong as real alternation ($p = 0.57$). However, it takes significantly longer to fully reduce the spike rate, as shown by the significant

difference between RA and FS before 200 ms after stimulus onset. **Bottom)** Average response of all face-selective single-units during flash suppression (blue) and real alternation (green) when perception shifts from place to face. Yellow indicates continued place perception. Spike rate increases compared to continuation ($p = 2.62 \times 10^{-40}$), but the increase is weaker compared to real alternation ($p = 6.01 \times 10^{-13}$). Horizontal bars indicate significance for real alternation (RA) and continuation of the initial stimulus compared (Con) to flash suppression (FS) ($p < 0.05$, two-sided permutation test with at least five consecutive significant time points). Vertical dotted line represents change from phase 1 to phase 2. Shading in (**A**–**C**) represents standard error (N = number of trials per condition). **D** Z-normalized net spike rate of all face-selective single-units during phase 2. Vertical dotted lines separate paradigms. Each column represents the z-normalized net spike rate in a 10 ms bin, thereby visualizing the first 300 ms of phase 2 per paradigm. Each row represents a separate single unit. Horizontal dotted lines separate arrays (A2 to A5 from top to bottom). The bar plot shows the average z-normalized net spike rate for all single-units over the first 300 ms. Pictures visualize the images shown to the nondominant (left) and dominant (right) eyes. Red bars below indicate perceptually suppressed stimuli. The face and place image in (**A** and **D**) are stock photos (face: iStock.com/Djavan Rodriguez; place: iStock.com/Iakov Kalinin).

## Binocular rivalry

During binocular rivalry, the same 2 images of a face and place were stationary presented to the two eyes for a prolonged duration, leading to alternating perception between both images. We recorded 78 visually responsive single units from five arrays, 35 of which were face-selective and 12 of which were place-selective (Table S8). Figure 6A

illustrates an example of both a face- and a place-selective neuron under rivalrous conditions. Both showed increased spiking activity prior to the self-reported perceptual emergence of their preferred stimulus. To assess population responses during perceptual transitions, we trained a decoder on all responsive units per array. Similarly to the flash suppression experiment, the decoder estimated the

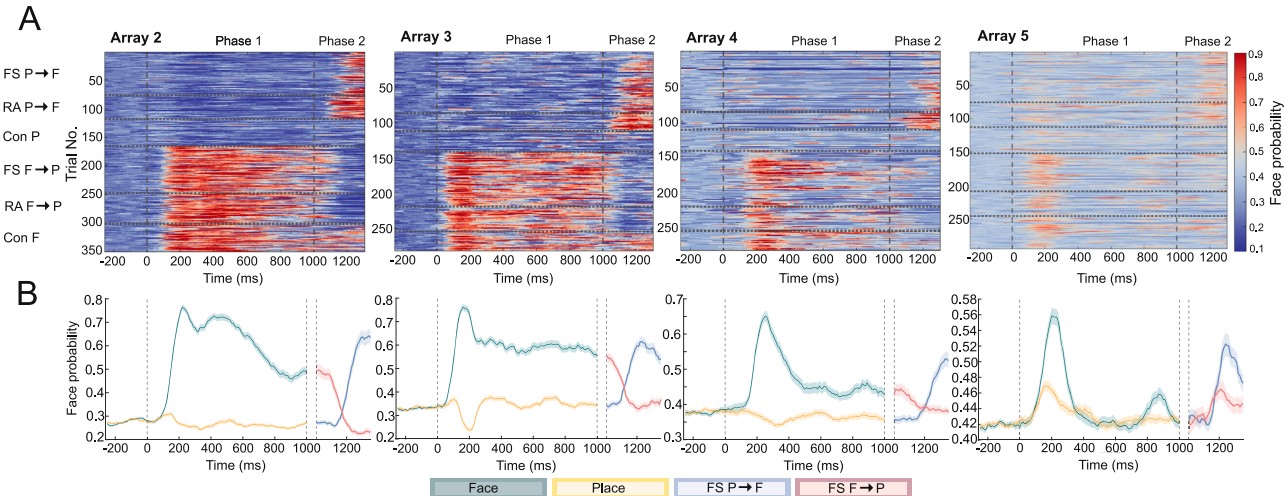

**Fig. 5 | Flash suppression: linear decoder. A** Bin-by-bin classification per trial for face and place using a logistic regression with LOO cross-validation. Horizontal dotted lines separate conditions. During the first 1000 ms, a place was presented in the first 3 conditions and a face in the last 3 conditions. FS = Flash Suppression, RA = Real Alternation, and Con = Continuation of the initial stimulus. Vertical dotted lines separate fixation, phase 1, and phase 2 of the experiment. Colors indicate the probability that a face is presented during a certain bin. **B** Decoder face probability for both flash suppression paradigms (FS F→P and FS P→F). Lines and shading represent mean ± standard error from all decoders per paradigm ($N_{decoders}$ = number of trials per paradigm). During phase 1, the green and yellow plots are the average prediction for a monocularly presented face and place, respectively. During phase 2, blue and red indicates face and place perception during flash suppression, respectively. Double-dotted vertical lines separate phase 1 from phase 2. For (**A** and **B**), x-axis represents the middle of the 100 ms bin used for decoding.

likelihood that a face or place was presented over time. Patient 1 failed to differentiate between the two stimuli during rivalrous conditions. Therefore, we do not present this patient's data in relation to the perceptual report. In arrays A2, A3, and A4, but not A5, the decoder signaled the identity of the upcoming percept approximately 1500 ms before the button press (Fig. 6B, C).

Next, we examined the decoder's predictions after stimulus onset, comparing predictions based on the first reported percept during rivalry. In all arrays, the decoders asserted significantly higher probabilities to a face being perceived on trials where the face was the initial percept, compared to trials where the place was perceived first (A2: $p = 8.94 \times 10^{-12}$; A3: $p = 0.042$; A4: $p = 0.029$; A5 $p = 0.030$, one-sided independent samples t-test comparing face probability within the response window, Fig. S23). When correlating the first reported image to the eye where the stimulus was presented in, we observed a strong eye dominance effect in patient 2 (array 2), where the first reported image matched the stimulus presented to the dominant eye in 95% of trials. In patients 3 (arrays 3 and 4) and 4 (array 5), the dominant eye stimulus was the first perceived image in 62% and 75% of trials, respectively. Since the sensory-dominant eye tends to gain perceptual dominance during extended periods of binocular rivalry[18], we further investigated whether the decoder consistently assigned higher probabilities to the dominant eye stimulus during prolonged rivalry. To test this, we compared the predicted face probabilities across all non-overlapping 150 ms bins from all rivalrous trials. We compared trials where the face was presented to the dominant eye (Face Dom Rivalry) to trials where the place was presented to the dominant eye (Place Dom Rivalry). Decoders asserted significantly higher probabilities to the stimulus that was presented to the dominant eye, except in A4 (Figs. 7B and S25, A2: $p = 1.02 \times 10^{-85}$; A3: $p = 8.32 \times 10^{-8}$; A4: $p = 0.60$; A5: $p = 5.59 \times 10^{-67}$; $A_{2-4}$ combined: $p = 4.97 \times 10^{-97}$, two-sided independent samples t-test). Notably, this effect was also evident in A1, where no dominant eye was identified during b-CFS, yet the face was more likely to be classified as the percept if it was presented to the right eye ($p = 2.81 \times 10^{-99}$, two-sided independent samples t-test).

## Discussion

We investigated the relationship between visual awareness and neural activity in the human lateral occipital complex using a range of perceptual paradigms. Our main findings are summarized in Fig. S26. Neural activity at the single-neuron level was closely tied to visual awareness. Decoders more accurately classified images when they were consciously perceived during backward masking, and asserted higher probabilities to the consciously perceived stimulus during periods of dichoptic stimulus presentation in flash suppression and binocular rivalry. These results offer evidence for a strong association between visual awareness and neural activity in LO, highlighting that subjective phenomenology, rather than purely physical stimulus features, predominantly drives neural responses in this region.

We focused on between-category selectivity, particularly the differentiation between faces and bodies[12,15–17,19,20]. Using naturalistic images of a face and place in our binocular rivalry and flash suppression paradigms, we observed selective responses across all arrays. However, this selectivity should not be interpreted as genuine face or body selectivity, as it may have been driven by lower-level visual features. While all arrays responded stronger to non-scrambled images, only array 4—and to a lesser extent, array 5—demonstrated clear category selectivity for bodies and faces. Consequently, the categorical stimuli used in backward masking were optimal only for array 4 and likely suboptimal for the other arrays. Although we were unable to perform flash suppression in array 1, and its activity related to the actual percept was uninterpretable during binocular rivalry, it nonetheless offered valuable insights. Analysis of its response properties indicated that array 1, despite showing a stronger response to non-scrambled images, was positioned in a distinct, more inferoposterior region of LO compared to the other arrays, reflecting early visual responses (Fig. 1A)[21]. This likely explains its strong responsiveness to the masking Mondrians and the inability to reliably decode the perceived stimulus category during backward masking.

Current understanding of the role of LO in the generation of conscious perception in humans has exclusively been derived from techniques that measure the compounded response of thousands of neurons, such as fMRI[22–24], EEG (scalp EEG[25–30], electrocorticography[31–33], intracranial EEG[24,34]), magnetoencephalography (MEG)[24,28,35,36], and event-related optical signal[30]. A recent meta-analysis of 54 fMRI studies concluded that the inferior division of LO is involved in conscious processing, while the superior division is associated with unconscious processing[23]. The most consistent visually evoked potential in EEG and

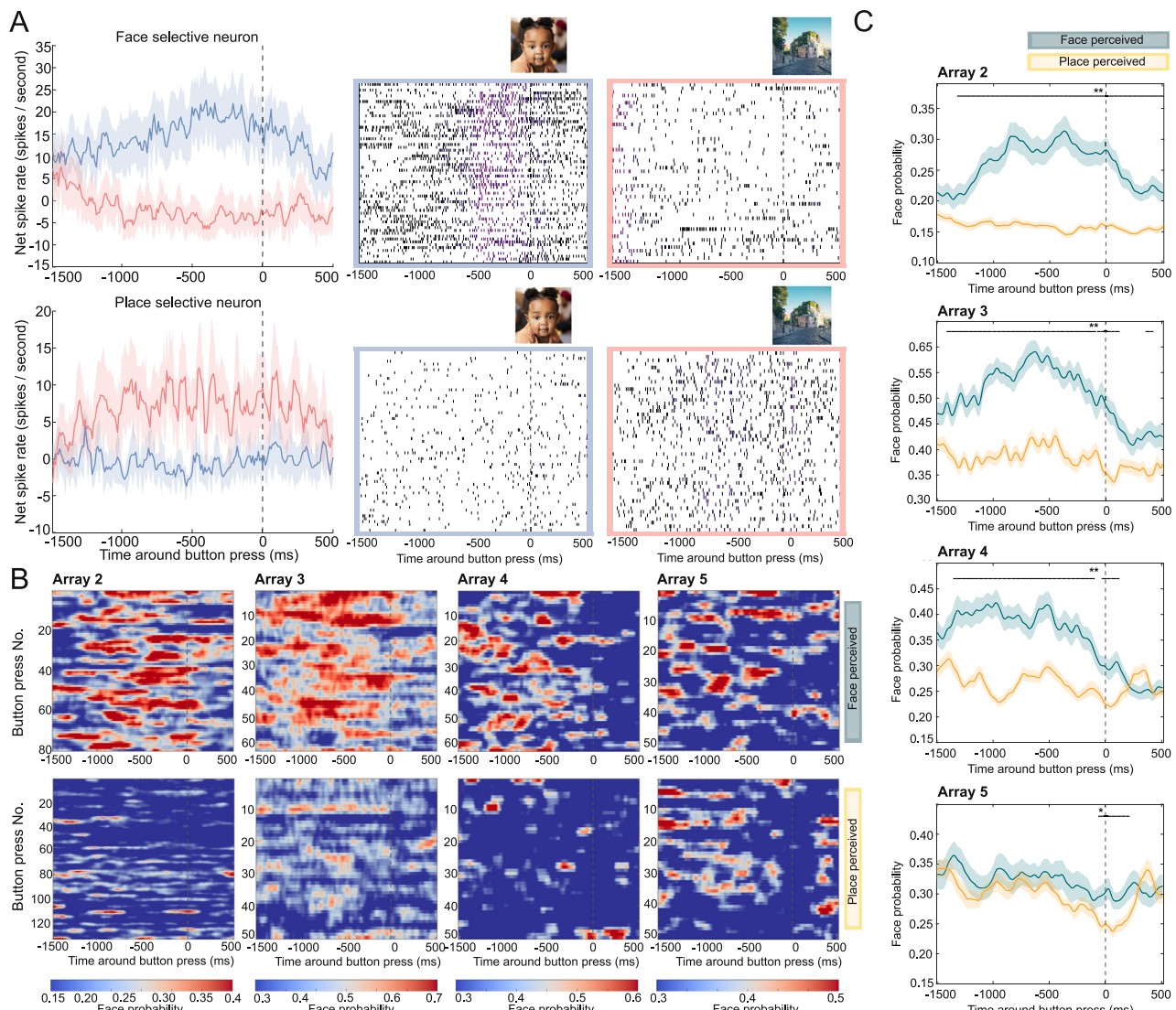

**Fig. 6 | Binocular rivalry. A** Example neurons (array 3) for self-reported perception (button press). Average net spike rate and raster plot of a face (top) and place (bottom) selective single unit around self-reported face **(blue)** and place **(red)** perception during rivalrous conditions. Dotted vertical lines indicate the button press signaling the report of perceptual change. Each horizontal line on the raster plots represents a separate trial. The face and place image are stock photos (face: iStock.com/Djavan Rodriguez; place: iStock.com/Iakov Kalinin). **B** Decoder face probability around self-reported perceptual changes of a face (top) and place (bottom). Each row represents a different instance where patients signaled a new percept. A Gaussian filter was applied for visualization ($\sigma = 1$). **C** Average decoder face probability during rivalrous trials when a face (green) or place (yellow) is reported. Horizontal bars represent significance (* $p < 0.05$; ** $p < 0.01$, one-sided permutation test with at least five consecutive significant time points). For all arrays, face probability was higher in the 1500 leading up to a face report compared to a place report (A2: $p = 5.17 \times 10^{-50}$, A3: $p = 1.34 \times 10^{-22}$, A4: $p = 1.74 \times 10^{-29}$, A5: $p = 0.005$; one-sided independent samples $t$-test). In (**A** and **C**), shading represents standard error ($N$ = number of button presses).

MEG that is associated with stimulus consciousness, irrespective of the report, is a negative ERP deflection over posterior and lateral occipital cortex after 100–300 ms, better known as Visual Awareness Negativity (VAN)[27,28,35–38]. Our decoding accuracy peaked 150–300 ms after stimulus onset, in agreement with literature[39]. Only if a stimulus enters awareness, a sustained activity in LO is observed in the temporal window of VAN, even outlasting the duration of a target stimulus, and its activity is predictive for later activity in striate and other extra-striate visual areas during conscious processing, suggesting a crucial role for LO in the generation of conscious visual perception[30,31]. LO activity increase appears to be the earliest predictor for perceptual reversal during binocular rivalry, yet the reversal occurs only when parietal and late visual (i.e., temporal lobe) sources become active as well[36]. Interestingly, we observed significant stimulus decoding ±1.5 s before self-reported perceptual changes. This is half a second faster than, to our

knowledge, the only single neuron recordings in human PFC that reported an increase in spike rate 1 s before perceptual changes[9]. This difference might reflect individual variability or differences in task instructions, so should not be taken as proof that occipital areas are responsible for initiating perceptual shifts.

To date, rare electrophysiological studies in humans on visual awareness at the single- and multiunit level primarily targeted frontotemporal regions such as the MTL and prefrontal cortex (PFC) due to their accessibility for depth electrode placement[2,3,9,40]. In the MTL, these studies often focus on concept cells, specialized neurons that respond to complex, high-level visual or memory representations of people and objects[2,3,9,40,41]. Activity within MTL subregions, such as the fusiform face area or parahippocampal place area, correlates with visual awareness[3,22]. However, only anterior, but not posterior MTL subregions differentiate perceived from nonperceived stimuli[40]. In

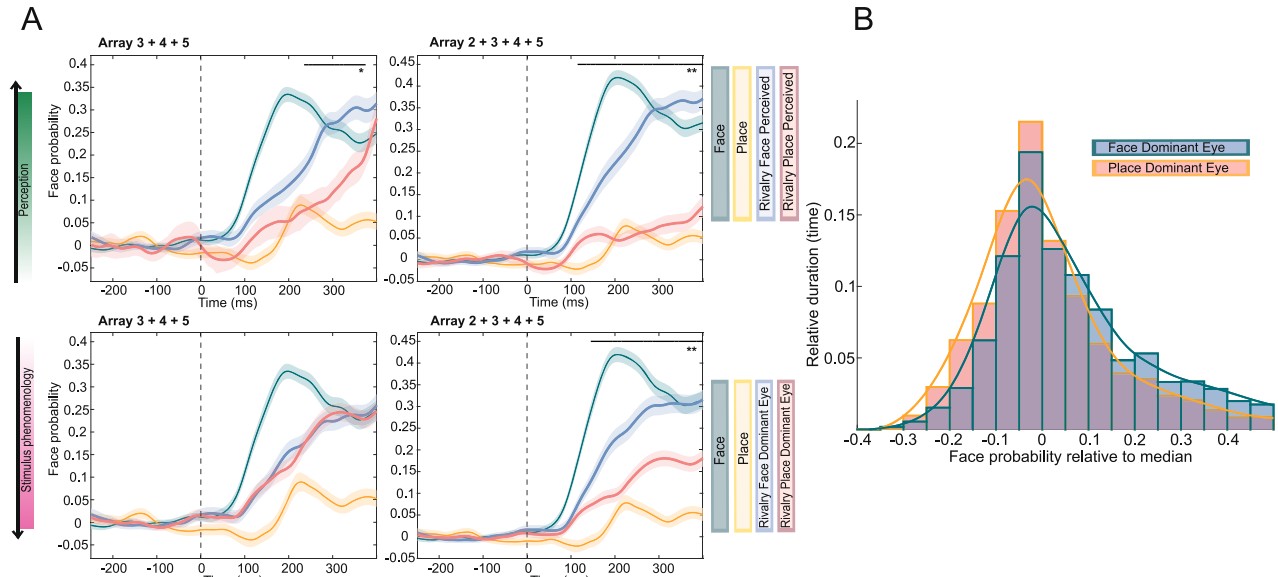

**Fig. 7 | Binocular rivalry: decoder face probability in relation to stimulus phenomenology and perception. A** Average decoded face probability (left: arrays 3, 4, and 5; right: array 2, 3, 4, and 5) after stimulus onset for non-rivalrous and rivalrous stimulus presentation, grouped by the first perceived stimulus (above) by the stimulus presented to the dominant eye (below). When perception is not taken into account, there is no significant difference between rivalrous conditions when excluding array 2 ($p = 0.639$; one-sided independent samples $t$-test comparing face (blue) vs. place (red) presented in the dominant eye 200–300 ms post-stimulus onset; including array 2: $p = 5.19 \times 10^{-5}$). However, when the face is perceived instead of the place during rivalry, the decoder assigns significantly higher face probabilities (excluding array 2: $p = 0.018$; including array 2: $p = 3.50 \times 10^{-10}$). This highlights a stronger distinction between rivalrous conditions when accounting for

perception. The $x$-axis time represent the middle of a 100 ms bin used for decoding. Vertical dotted line marks stimulus onset. Horizontal bar represents significance (blue graph > red graph, one-sided permutation test, * $p < 0.05$, ** $p < 0.001$). Shading represents standard error. A subdivision was made with and without array 2 due to the very strong eye dominance effect observed in array 2. Before grouping, mean baseline face probability was subtracted for all conditions per array. The same plots for each array separately are presented in Supplementary Fig. S23. **B** Probability density function of decoder results for arrays 2, 3, 4, and 5, illustrating that face perception probability is higher when the face stimulus is presented to the dominant eye (blue) compared to the nondominant eye (red). To facilitate comparisons across arrays, the median decoder probability was subtracted for each array individually.

---

contrast to human research, macaques have been extensively used to investigate the correspondence between neural activity and perceptual report along the ventral visual stream[6,8,42,43]. The proportion of neurons following the perceptual report increases along the visual processing hierarchy and neuronal activity only closely tracks perceptual reports within the inferior temporal cortex (ITC), with 60–90% of neurons modulating according to conscious perception[4,8,10,11]. This is further supported by correlational[44,45] and causal[46–48] techniques, suggesting that anterior ITC plays a critical role in conscious visual perception. Our single-unit recordings in human LO reveal parallels with macaque ITC[6–8,42]. First, in macaques, backward masking with brief delays shortens the selective responses of ITC neurons[7]. While our experiment was not designed to confirm or disprove this effect, we observed that selective responses were interrupted by the mask when the delay between target and mask was short (Fig. S10B–D). In contrast, nonselective neurons exhibited a single response (Fig. S10A). These findings in human LO, like those in macaque ITC, support the interruption theory of masking over the integration theory[7,49]. Second, 64–69% of LO neurons modulated their activity in accordance with the percept during flash suppression. However, this might be an underestimation, as we only tested a single stimulus pair[43]. The number of perception-modulating neurons could vary depending on the specific face-place pair used. A neuron might track perceptual changes for one such pair but not for another, even if its responses to the nonrivalrous versions of both stimuli are identical[43]. Moreover, the use of a single stimulus pair limited our ability to investigated if LO activity yields sufficient information to decode perceptually suppressed stimuli from each other[11]. Despite all patients confirming the absence of residual perception of the suppressed stimulus, nearly all LO neurons (> 94%) responded less strongly to flash-suppressed stimuli compared to real alternation (Fig. S18C), which was also reflected by weaker prediction

probabilities by the decoders, indicating a graded response consistent with fMRI studies of LO and single-neuron recordings in non-human primate ITC[8,11,50,51]. This suggests that some LO neurons encode sensory input rather than perceptual awareness, which is further supported by the observation during binocular rivalry that certain LO neurons responded exclusively to rivalrous conditions, while others tracked the perceived stimulus (Fig. S27). In contrast, human MTL neurons exhibit a more binary response, with approximately two-thirds following the percept in an all-or-none manner[2–4,9,40,52]. Therefore, LO may function as an intermediary region between low- and high-level visual areas, contributing to the generation of conscious perception without the binary, all-or-none modulations characteristic of higher-order regions. This is in line with Fisch et al.[31] where low but significant gamma activity, which correlates well with spiking activity[53,54], was observed in LO during unperceived backward masking trials.

In our backward masking experiment, decoding accuracy improved with longer delays between the target stimulus and the mask. However, we were still able to decode nonperceived target stimuli at these longer delays. This could indicate either that LO retains sufficient information for decoding unperceived stimuli, or that patients occasionally misclassified perceived stimuli. While we cannot definitively distinguish between these two explanations, we consider the latter more likely. This is supported by the lack of reliable decoding at the shortest delays (≤ 32 ms), and by the fact that our patients with long-standing intractable epilepsy failed to achieve high classification accuracy even at the longest delays, unlike healthy subjects in our lab, who performed near-ceiling under the same conditions. A no-report paradigm, where patients are not required to report their perceptual experiences, might have been more suitable for these patients. Our flash suppression paradigm was conducted without perceptual reporting, but no-report paradigms also exist for backward masking[55]

and binocular rivalry, often relying on physiological measures of pupil size, optokinetic nystagmus, or position of fixation[11,56,57]. No-report paradigms might still be susceptible to attentional biases[58], which we attempted to mitigate by training decoders on the baseline interval; however, their main benefit resides in avoiding confounds introduced by reporting-related activity. Report-related activity may mainly affect frontal rather than occipital regions[55,57]. Therefore, while we believe a no-report paradigm could have avoided potential reporting effects, we believe report-related activity had minimal influence on our results.

Although LO is only one component of a broader network, our findings may offer valuable insights into how conscious visual percepts are generated within this larger system. Two prominent theories attempt to explain how specific brain areas and their interactions give rise to conscious awareness. The Global Neuronal Workspace Theory (GNWT) postulates that information in sensory areas remains unconscious until it is broadcast globally by prefrontal cortex, engaging a distributed network of prefrontal, parietal, and high-level sensory cortical areas[59–61]. While prefrontal cortex appears to transiently signal the onset of a new percept, but not the offset, this observation simultaneously supports and challenges the GNWT[9,24,62,63]. In contrary, the Integrated Information Theory (IIT) posits that consciousness emerges from a high degree of information integration within a network, particularly in a temporo-parietal-occipital hot zone that includes LO[59,64]. According to IIT, this region generates maximal integrated information (phi), supporting conscious visual awareness through complex, reciprocal interactions with other brain regions that create unique cause-effect dynamics. While several studies have reported sustained activity in this posterior hot zone for perceived stimuli[24,31,63], others have not[34]. More recently, one study failed to find sustained synchronization in the posterior cortex during visual awareness, raising questions about the IIT framework[24]. In our flash suppression data, nearly all (> 99%) LFP recordings sites showed sustained gamma-band activity for the full 1000 ms duration of stimulus presentation (Fig. S22). However, only 7 out of 37 neurons (19%) maintained elevated spiking activity throughout this period. Linear decoders trained on SUA and LFP differentiated between both stimuli for the entirety of stimulus presentation in 3 out of 4 arrays (array 2, 3, and 4, but not array 5). This suggests that while population-level signals may reflect persistent stimulus-specific information, this information is carried by only a small subset of all selective neurons within LO. Lastly, perceived stimuli elicited 20–40% stronger responses than unperceived stimuli during backward masking, even earlier than the selectivity latency. Such divergence has also been observed in areas V1 and V4 in non-human primates, regardless of stimulus contrast[65,66]. Therefore, our results agree with a gradually increasing divergence in neural activity between perceived and unperceived stimuli during the feedforward propagation of information, in which there is increasing signal loss for unperceived stimuli over consecutive visual areas. This could be compatible with the hypothesis that a critical activity threshold must be met in the PFC to broadcast it globally[3,4,65].

Since we did not simultaneously record from PFC, and our analysis did not directly measure cause-effect structures from the neural activity patterns as predicted by IIT (e.g., as done by Haun et al.[33]), we cannot draw definitive conclusions in support of either theory. Moreover, while our decoding and single-unit analyses demonstrate that neuronal activity in LO is modulated by perceptual awareness, these findings neither establish causality nor confirm that LO is necessary for the generation of conscious perception[67]. Future research could delve deeper into the functional and anatomical connections between LO and higher-order areas, especially regions suggested by the GNWT, such as the PFC and MTL. Such studies would benefit from simultaneous recordings of spiking activity across these regions to better capture the dynamics of information flow underlying conscious perception.

## Methods

### Study approval

We obtained ethical approval (study number s53126) for conducting semi-chronic microelectrode recordings using the Utah array in patients undergoing invasive epilepsy monitoring. The study protocol was approved by the local ethical committee (Ethische Commissie Onderzoek UZ/KU Leuven). The study was carried out in accordance with the principles of the Declaration of Helsinki, the principles of good clinical practice, and all applicable regulatory requirements set by the Federal Agency for Medicines and Health Products.

To ensure safety and compliance, strict adherence to imposed safety measures was maintained. This included the use of case report forms and detailed reports on any (serious) adverse events. All data collected during the study were encrypted and stored at the University Hospitals Leuven. The study was thoroughly discussed with the patients during the pre-operative consultation, which took place at least six weeks before the surgery. The patients were informed about the risks associated with micro-electrode array implantation, such as infection and hemorrhage. No additional incisions were made for the purpose of this study. Written informed consent was obtained from the patients on the evening prior to their surgery.

### Clinical information

We obtained invasive intracranial recordings from the lateral occipital cortex in 4 refractory epilepsy patients. Patient 1 was a 24-year old male with no prior medical history besides epilepsy. Patient 2 was a 55-year old female with neurofibromatosis type 1 (NF1). She suffered from a left occipito-temporal brain hemorrhage caused by venous sinus thrombosis at the age of 35 (MNI coordinates: anterior border (−65, −45, −5), superior border (−54, −62, 6), posterior border (−53, −75, −2), and inferior border (−54, −60, −16)). She had no intracranial tumors. Patient 3 was a 58-year old female suffering from recurrent psychotic episodes. Patient 4 was a 29-year old male who underwent resective epilepsy surgery at the right temporoparietal junction at the age of 19 (MNI coordinates: anterior border (64, −42, 32), superior border (51, −53, 56), posterior border (55, −67, 42), and inferior border (60, −60, 27)). The pathology report after this resection mentioned chronic meningitis. After 1 year of seizure freedom, seizures recurred, and invasive intracranial recordings were performed as part of the diagnostic epilepsy surgery workup.

All patients were taking chronic anti-seizure medications: patient 1 took Brivaracetam 100 mg, Lacosamide 200 mg, and Oxcarbazepine 450 mg, all twice daily. Patient 2 was on Levetiracetam 2000 mg and Lamotrigine 225 mg, both twice daily. Patient 3 was on Levetiracetam 2000 mg, Diphantoine 100 mg, Clonazepam 1 mg, and Lacosamide 300 mg, all twice daily. Patient 4 was on Levetiracetam 2000 mg and Lacosamide 300 mg, both twice daily. During admission to the epilepsy ward, the anti-seizure medications was adjusted for all patients based on the number of detected seizures.

Target locations of intracranial electrodes were determined by the epileptologist in a multidisciplinary consensus meeting and were based on electroclinical findings and non-invasive multimodal imaging. Multi-electrode arrays were implanted via the craniotomy performed for the implantation of subdural grids, adjacent to the grids (Fig. S29). The arrays were removed during the same surgery for subdural grid removal, 14–16 days after implantation. Both insertion and removal of electrodes were done under general anesthesia. No new neurological deficits, nor implantation-related complications (infection, hemorrhage, wound problems) were recorded (Fig. S29).

Although the presumed epileptogenic zone encompassed the lateral occipital cortex based on multimodal preoperative assessment, no focal onset could be identified in some patients. In patient 1, seizures originated from the right amygdala, and a right amygdalo-hippocampectomy was performed a couple of months later, after which the patient was deemed seizure-free. In patient 2, invasive

recordings showed no clear focal onset. In patient 3, two arrays were inserted in lateral occipital cortex. Seizure onset was multifocal and no resection was proposed. In patient 4, seizures originated from the inferior border of the previous resection, and a more extensive resection was performed at the time of electrode removal (Figs. S28 and S29). In all patients, multi-unit spiking activity was analyzed to investigate to what extent neural activity could predict seizure onset[68].

## Invasive recordings

We recorded neural activity using a 96-channel array (Utah Array™– Blackrock Neurotech, USA) from day 1, the day after implantation, until the day before electrode removal. Arrays were inserted with a single pneumatic hit (20 psi) using an inserter wand provided by Blackrock Neurotech. We closed the dura over the array and placed the bone flap on top to secure the array and prevent array flotation. Reference wires were placed subdurally and ground wires were placed epidurally. The signal was digitally amplified using a Cereplex M head stage from Blackrock Neurotech, connected to a 128-channel neural signal processor (NeuroPort), and sampled at 30 kHz using Central Software. We applied a 750 Hz high-pass filter to isolate spiking activity, with the multi-unit detection threshold set at −3 standard deviations of the signal. Signal quality varied during the immediate postoperative period, leading to fluctuations in the number of responsive and selective channels across recording sessions. Spike sorting was performed offline (Offline Sorter 4, Plexon, TX).

## Imaging

T1-weighted images were acquired via a 3-T MR scanner (Achieva dStream, Philips Medical Systems, Best, the Netherlands) at the University Hospitals Leuven for presurgical planning. The imaging parameters were as follows: time of echo 3.1 ms, inversion time 900 ms, slice thickness 1.0 mm, 400 × 400 matrix size, and 283 coronal slices. Following electrode placement, a CT scan (Siemens) was performed with a slice thickness of 1 mm, a voltage of 120 kV, and a dose length product of 819 mGy.cm. The CT scan was used to identify the exact electrode locations and rule out any hemorrhage. This postoperative CT was later fused with a postoperative MRI performed months after surgery to rule out structural alterations due to electrode insertion. No structural alterations (gliosis, ischemia, hemosiderin) were observed at the implantation sites (Fig. S29)[15].

Imaging preprocessing was done with SPM12 software (Wellcome Department of Cognitive Neurology, London, UK) running on MATLAB (MathWorks, Natick, MA, USA). The preprocessing steps included (1) setting the anterior commissure as the origin for both the MRI and CT scan, (2) realigning the images, (3) co-registering the MRI and CT images, and (4) warping the MRI and coregistered CT to a common high-resolution brain atlas (MNI152). FreeSurfer was used to generate cortical 3D renderings of the subjects' pial surfaces[69]. The precise electrode location on the cortical surface and the MNI coordinates were determined via iElectrodes Software[70].

## Stimulus presentation

We conducted experiments in a dimmed hospital room using custom-built software for stimulus presentation. Two different setups were used: setup 1 consisted of a 60 Hz DELL-P2418HZM LED monitor, where patients were seated 60 cm away from the screen (1 pixel = 0.026°); setup 2 consisted of a custom-built mirror stereoscope where images were presented separately to the right and left eyes on two 120 Hz LCD monitors (Benq XL2420T) with the use of customized mirrors at a viewing distance of 56 cm (1 pixel = 0.028°). In both setups, fixation was monitored using continuous eye-movement tracking (setup 1–left or right eye, 1000 Hz, Eyelink 1000 plus in head free-to-move mode; setup 2–left eye, 120 Hz, ISCAN, MA). In both setups, patients were instructed to maintain fixation on a small red square

(0.2 × 0.2°) positioned at the center of the display. If the participants' gaze deviated beyond the electronically predefined 3° by 3° fixation window, the trial was aborted. To synchronize data streams, a photodiode was attached to the upper left corner of the screen, which detected a bright flash coinciding with the first frame of the stimulus. This flash was invisible to the patients. To accurately record baseline spiking activity, an intertrial interval of 100 or 150 ms was introduced (i.e., the offset of the stimulus and the onset of fixation for the next trial). Each trial was preceded by a 300 ms fixation period.

## Stimulus sets and experimental designs

**LO localizer stimuli: scrambled versus non-scrambled.** We used 21 neutral naturalistic chromatic images, as well as their achromatic counterparts for a total of 42 images, of animals and humans (including bodies and faces) from the International Affective Picture System (IAPS) database[71]. Luminance was equalized across all chromatic stimuli via the SHINE toolbox and luminance was maintained across each chromatic-achromatic image pair[72]. Each of those 42 images were scrambled for a total of 84 images in the stimulus set (scrambling block size 0.1°). Stimuli measured 4 × 4° and were presented for 300 ms. Patients performed a distractor task, responding via button press when a cross appeared at the fixation marker. This experiment was performed on days 15, 2, 5, and 1 in patients 1 through 4, respectively. During the first week, the signal quality from array 3 in patient 3 was poor, while array 4 had good signal quality. As a result, the task was repeated on day 12 for array 3.

**Category selectivity stimuli.** This stimulus set consisted of 100 achromatic stimuli across 10 categories[73–75]. The categories included mammals, birds, fruits, bodies (human–monkey), faces (human–monkey), objects (matched in aspect ratio of images from the human class "objects-human" and monkey class" objects-monkey", respectively), and sculptures. For this analysis, we focused only on the human face, human body, objects-human, and fruit category classes given the fact that these stimuli were used in the backward masking experiment. The mean luminance values were equalized across all categories. The mean vertical and horizontal extent of the images were 6.0° and 4.6°, respectively. The stimuli were presented on a uniform gray background for 250 ms with an intertrial interval of 150 ms. Patients performed a passive fixation task. This experiment was performed on days 16, 3, 3, and 1 in patients 1 to 4, respectively. The task was repeated in patient 3 (array 3) on day 14 because of poor signal quality on day 3.

**Eye dominance testing: breakage of continuous flash suppression (b-CFS).** Eye dominance was assessed via setup 2 by measuring the latency of suppression breakage in a continuous flash suppression paradigm (Fig. 2A)[76]. Dynamic noise, composed of overlapping and differently sized grayscale squares– referred to as Mondrians–was presented to one eye at a frequency of 10 Hz. Simultaneously, a black arrow pointing either left or right was displayed to the other eye, positioned above, at, or below the fixation marker on a gray background. This variable arrow position prevented patients from selectively attending to a single location. The contrast of the arrows linearly increased from 0 to 100% over a 10-s period, whereas the contrast of the Mondrians linearly decreased from 100% to 0% over the same period. To facilitate binocular image fusion, a frame of black and white circles was presented binocularly. Patients were instructed to press a button with their left or right hand, indicating the direction of the arrow, to determine the latency of suppression breakage. Each trial ended upon suppression breakage or after 10 s if suppression did not break. Both eyes were tested simultaneously in random order. The dominant eye was identified as the one with the shortest suppression breakage latency when presented with the arrows. A two-sided independent samples $t$-test was used to test for significance. The

binocularly presented frame, including the Mondrians, measured 8° by 8°, while the arrows were 2° in size.

**Backward masking.** Backward masking was conducted via setup 1 (Fig. 2B). A set of target stimuli was presented for 16 ms. This was followed by a variable delay during which a gray background was displayed, lasting 16, 32, 50, 66, 83, 100, 116, 132, 150, 166, 183, or 200 ms. Subsequently, a static Mondrian mask was shown for 250 ms. If the Mondrian was presented immediately after the target stimulus, the delay was defined as 16 ms. Mondrians measuring 8° × 8° were randomly generated for each trial. Target stimuli could be categorical images (human faces, human bodies, fruits, and objects-human) from the category-selectivity experiment (performed in all patients) or scrambled and non-scrambled images from the scrambled vs. non-scrambled experiment (performed only in patient 2). Following the Mondrian presentation, patients had 1500 ms to verbally indicate the category of the target (in case of the categorical stimuli) or whether they saw a non-scrambled image. Responses were scored by the investigator (M.V.). Patients were instructed not to guess in case of uncertainty. A stimulus was deemed unperceived in case the target stimulus was categorized wrongly, or if the patient did not make a verbal report. In a separate recording session on the same day, isolated responses to the categorical target stimuli (presented for 16 ms without the masking stimulus) and to the Mondrians (presented for 250 ms) were recorded. These additional data were obtained from patients 2, 3, and 4 and provided a baseline for neural responses (separate from the backward masking task). The experiment with categorical images was performed on days 15, 11, 6, and 6 in patients 1 to 4, respectively. The experiment with scrambled and non-scrambled images was performed on day 13 in patient 2.

**Flash suppression.** Flash suppression was performed via setup 2 (Fig. 2C). Each trial began with the monocular presentation of a naturalistic image of a place or a face to the nondominant eye. Simultaneously, a gray background was presented to the dominant eye. After 1000 ms, the perceived image was manipulated in one of three ways: (1) flash suppression (FS) in which the perception of the initial image was suppressed by the sudden presentation of a new image to the dominant eye, while the initial image remained visible to the nondominant eye; (2) real alternation (RA) in which the new stimulus was introduced to both the dominant and nondominant eye; and (3) continuation (Con) of the initial stimulus to the nondominant eye, with no stimulus change, thereby not inducing any perceptual shift. Trials from each condition were presented in a randomized order. The second phase of each trial lasted 400 ms. Before the main experiment, each subject underwent preliminary testing to ensure the effectiveness of flash suppression. This ensured that there was no residual perception of the suppressed image, allowing the main experiment to proceed without requiring patients to indicate the perceived image during trials. The same frame used in the ocular dominance testing was presented binocularly during each trial to optimize binocular fusion. Stimuli measured 8° by 8°. The experiment was performed on days 12, 10, and 13 for patients 2, 3, and 4, respectively.

**Binocular rivalry.** The same two stimuli used in the flash suppression experiment—a naturalistic image of a face and place—were used (Fig. 2D). Each trial lasted 24 s for patient 1. Because it was difficult to maintain fixation for such an extended period of time, we decreased the trial duration to 10 s for patients 2, 3, and 4. To identify selective neurons that differentiate between both images, three conditions were included: (1) continuous binocular presentation of the face; (2) continuous binocular presentation of the place; and (3) alternation between the face and place at a frequency of 1 Hz with binocularly presented stimuli. Conditions 1 and 2 were implemented after a preliminary data analysis in patient 1, and were therefore obtained only in

patients 2, 3, and 4. In conditions 4 and 5, the face and place images were presented to opposite eyes to induce binocular rivalry. The distribution of images between the dominant and nondominant eye was counterbalanced across trials for the rivalrous conditions. All 5 conditions were designed to elicit and identify neuronal selectivity for the two distinct visual stimuli. This approach allowed us to examine how neurons differentiate between the face and place images under continuous presentation, alternating presentation, and binocular rivalry. The experiment was performed on days 14, 10, 10, and 13 for patients 1 to 4, respectively.

## Analysis

**Data normalization.** We analyzed data using custom-written MATLAB R2023b (MathWorks, Natick, MA, USA) scripts. Net spike firing rates were calculated from raw single- and multiunit firing rates by subtracting baseline activity (300 ms interval before stimulus onset) from the entire trial, binned in 10 ms intervals. Analyses were restricted to visually responsive sites. To assess the responsiveness of each recorded site, we performed a two-sided dependent samples $t$-test to compare the average spike rates during the response windows to baseline (with an average spike rate of 0) for each stimulus category/condition separately. This approach ensures that sites with a strong preference for a specific category or stimulus are not overlooked by a broad test across all stimuli. For each experiment, a site was considered visually responsive if any condition showed a significant difference from baseline ($p < 0.005$) in one of the following 100 ms intervals after stimulus onset: [100–200], [150–250], [200–300], or [250–350]. These multiple intervals account for variations in response and selectivity latencies across arrays. All subsequent analyses utilized baseline-subtracted average net firing rates or z-normalized net firing rates. Z-normalization was performed for each 10 ms-binned multiunit or single-unit recording as follows:

$$Z(i) = \frac{R(i) - R_{mean}}{R_{std}}$$

where $Z(i)$ corresponds to the Z-normalized net spike rate for a specific 10 ms bin, $R(i)$ corresponds to the average net response from the same 10 ms bin, $R_{mean}$ is the average net spike rate over all trials in the response window (*see below*), and $R_{std}$ is the standard deviation in the response window. This method ensures that the firing rates are standardized, allowing for consistent comparisons across different conditions and sites. The reported Z-score reflects the Z-normalized net spike rate within the response window, with an average Z-score of zero across all conditions for each single- or multiunit.

**Local field potentials and spectral power.** Local field potentials (LFPs) were preprocessed to reduce line noise and extract relevant frequency components. Data were filtered with a combined spectral and spatial filter[77], which can eliminate artifacts while minimizing the deleterious effects on non-artifact components. We applied zero-phase, second-order IIR notch filters at 50, 100, and 150 Hz to remove power line artifacts. Subsequently, the signal was bandpass filtered between 1 and 450 Hz using a linear-phase finite impulse response (FIR) filter applied in both forward and reverse directions. For every trial, the time-frequency power spectrum was calculated using Morlet's wavelet analysis techniques[78], with a spectrotemporal resolution of 7. Spectral power was normalized per trial by dividing the power trace per frequency by the average power for each frequency in the 200 ms interval before stimulus onset, resulting in an average normalized baseline activity of 1 (for each frequency). The 60–120 Hz frequencies were averaged as a measure of gamma-band activity.

**Response and selectivity latency.** Response latency was calculated on the net multiunit firing rates and defined as the first of 3 consecutive

10 ms bins in which the net response per bin increased compared to the previous bin, and was greater than 2 standard deviations above the baseline firing rate. To account for anticipation, which was particularly evident in patient 2 (array 2), net spike rate had to increase by >50% relative to the previous bin and by 5 spikes per second in absolute terms. Response latencies were calculated for both the category-selectivity and scrambled vs. non-scrambled experiment. Selectivity latency was calculated only for the scrambled vs. non-scrambled experiment and was defined as the first of 3 consecutive bins in which $\mu_{non-scrambled.} > \mu_{scrambled}$ ($p < 0.05$, one-sided independent samples $t$-test), with $\mu$ representing the mean response per bin. The middle of the bin was used for reporting latencies.

**D-prime index.** The d-prime index or d′ quantifies the discriminability between groups by considering the mean responses to corresponding groups as well as the variability of the responses to the individual images of a group[79]. The d′ indices were computed for various pairs of groups using net firing rates. We calculated d′ for each array within the interval defined by the fastest selectivity latency and selectivity latency + 150 ms as follows:

$$d' = \frac{\mu_{group1} - \mu_{group2}}{\sqrt{\frac{\sigma^2_{group1} + \sigma^2_{group2}}{2}}}$$

where $\mu_{group1}$ and $\mu_{group2}$ are the mean firing rates and where $\sigma_{group1}$ and $\sigma_{group2}$ are the variances in firing rates. The respective applied intervals for arrays 1 to 5 were [100 250], [130 280], [130 280], [200 350], and [140 290]. These intervals are specific for each array and are referred to as the response windows. A positive d′ indicates that the net response of group 1 > group 2. The significance of a single- or multiunit d′ was assessed by comparing the obtained values to a null distribution of values generated by randomly permuting class labels across trials (1000 iterations).

**Category selectivity.** We calculated d′ values between 3 groups from the category selectivity experiment (human faces, human bodies, and objects in which objects-human, objects-monkey, fruits, and sculptures were grouped). One-way ANOVA was performed to determine if a unit was selective for one or more categories within the response window ($p < 0.05$). We then performed Tukey's Honest Significant Difference (HSD) test to assess significance between individual groups, e.g., face vs body. A unit was face or body-selective if $\mu_{face(body)} > \mu_{object}$ and $\mu_{face(body)} > \mu_{body(face)}$ ($p < 0.05$). A unit was object-selective if $\mu_{object} > \mu_{face}$ and $\mu_{object} > \mu_{body}$ ($p < 0.05$). A unit was both face and body-selective if $\mu_{face}$ and $\mu_{body} > \mu_{object}$ ($p < 0.05$) and $\mu_{face}$ vs $\mu_{body}$ ($p > 0.05$).

**Silverman's test for bimodal responses during backward masking[80].** We identified both responsive and selective units for each array. A single unit was considered selective for a specific class if the average net response within the response window was greater than the response to all other classes for delays ≥150 ms, as the Mondrian is unlikely to interfere with neural responses at such long delays ($p < 0.01$; one-sided permutation test). The analysis was conducted per delay, per array, and separately for selective and responsive units. We grouped all z-normalized spike traces per trial (corresponding to a certain delay) for both selective and responsive units, averaging the traces in random groups of three to reduce noise. Each trace was only used once, so this averaging reduced the number of traces available for further analysis by a factor of 3. The decision to average per three traces was arbitrary, but we felt that this provided a good trade-off between noise reduction, while still maintaining enough averaged traces for further analysis. The averaged traces were then smoothed using a 50 ms moving average. Next, we identified the largest response

peaks−with a minimum of 0 and maximum of 2−after stimulus onset within each smoothed average trace, based on the following criteria: (1) a minimum distance of 30 ms between peaks; and (2) a peak spike rate exceeding 30% of the highest average spike rate across all delays for the tested class. We then bootstrapped all timepoints corresponding to the identified peaks 1000 times with replacement and estimated the kernel density (KDE) of each bootstrap sample (bandwidth = 2). Peaks in the estimated kernel densities were identified, with at least one peak always present. A second peak was recognized if (1) its height exceeded 30% of the first peak and (2) neighboring valleys were at least 30 ms away on both sides. The $p$-value was calculated as follows:

$$p = \frac{1}{1000} \sum_{i=1}^{1000} 1(pk(B_i) = 1)$$

where $pk(B_i)$ represents the number of identified peaks in the KDE of the $i$-th bootstrap sample B. This method effectively identifies and distinguishes peaks when a strong neural response is present. However, in the absence of a significant neural response, it is susceptible to Type I errors (Fig. S30). This occurs because fewer peaks are detected within the neural traces, leading to a more heterogeneous KDE, which can erroneously suggest the presence of multiple peaks.

**Quantification of signal loss during backward masking for unperceived stimuli.** We quantified the signal loss for unperceived stimuli in response to the stimulus class eliciting the strongest neural response via the following formula:

$$\text{Signal loss} = \left(1 - \frac{\text{Activity}_{non-perceived}}{\text{Activity}_{perceived}}\right) * 100\%$$

where Activity$_{non-perceived}$ and Activity$_{perceived}$ are the z-normalized responses within the 150 ms interval following the response latency to all nonperceived and perceived stimuli, subtracted by the mean baseline z-normalized response for all nonperceived and perceived stimuli, respectively. We chose to use the response latency rather than the selectivity latency to capture any changes that occur before selectivity fully emerges. The latency intervals used for arrays 1 through 5 were as follows: [60, 210], [80, 230], [130, 280], [200, 350], and [130, 280], respectively. Significance was assessed using a two-sided independent samples $t$-test.

**Response types during flash suppression.** We first identified single units and LFP channels selective to the face image ($p < 0.05$, two-sided independent samples $t$-test comparing face vs place within the response window). To evaluate the effect of flash suppression when perception shifts from the preferred (face) to the non-preferred stimulus (place), it is crucial that (1) a neuron increases its spike rate during the entire duration of the preferred stimulus presentation, or (2) a neuron decreases its spike rate when presented with the non-preferred stimulus. On the basis of these criteria, we identified (1) sustained responders and (2) inhibitory responders. A neuron or LFP channel was classified as a sustained responder if the mean net firing rate or gamma activity (60–120 Hz) during the last 200 ms of the first phase of continued face presentation ($\mu_{800-1000}$) was greater than the mean net firing rate during the 300 ms fixation period ($\mu_{800-1000} > \mu_{fixation}$) ($p < 0.05$, one-sided dependent samples $t$-test). A neuron was classified as an inhibitory responder if the mean net firing rate after real alternation from face to place (RA F→P) was lower than the mean net firing rate after continuation of the preferred stimulus (Con F) ($p < 0.05$, one-sided independent samples $t$-test) within the response window. Notably, most sustained responders are also likely to be classified as inhibitory responders. Therefore, an inhibitory

responder should not be interpreted as a unit actively inhibiting its activity in response to the nonpreferred stimulus.

**Identification of perception tracking neurons.** Neurons tracking perception were identified during flash suppression when perception shifted either from the preferred to the nonpreferred image or vice versa. We assessed (1) single units that selectively responded to face or place images within the response window ($p < 0.05$, two-sided independent samples $t$-test comparing face vs place within response window) and showed significant activity compared to baseline after real alternation from the nonpreferred to the preferred image ($p < 0.05$, one-sided dependent samples $t$-test). A neuron was classified as following perception if its response after flash suppression from the nonpreferred to the preferred stimulus exceeded its response during the continued presentation of the nonpreferred stimulus ($p < 0.05$, one-sided permutation test). Additionally, we analyzed (2) the sustained and inhibitory responders, categorizing them as percept-following if their spike rates within the response window were lower than those during the continued presentation of the preferred stimulus after a perceptual shift through flash suppression and real alternation from the preferred (face) to the nonpreferred (place) stimulus ($p < 0.05$, one-sided permutation test). Some selective, sustained, or inhibitory neurons did not meet the criteria to be evaluated for percept-following behavior.

**Decoding class information.** We used linear decoders to estimate the amount of information present in the neural population during subliminal presentation of different category classes. This analysis was conducted on a trial-by-trial basis, for each recording session separately using every visually responsive site, to prevent bias toward a specific category. To quantify neural information in a time-resolved manner, firing rates were calculated within 100 ms bins with a 10 ms sliding window. For each time bin, we trained a logistic regression classifier with L1 regularization (LASSO) and C = 0.1 in MATLAB. The benefit of L1 over L2 regularization (ridge regression) is that it deletes unnecessary multi- or single-unit activation sites and shrinks its components to exactly zero[81]. Before classification, net spike counts were z-normalized to ensure that classification was not influenced by the absolute magnitude of the responses.

To address class imbalance, we adjusted the logistic regression loss function weights inversely proportional to the class frequencies in the input data. For the multiclass problems (e.g., in the category-selectivity experiment and backward masking of categorical stimuli), we employed a one-vs-rest approach, training the classifier to distinguish between each category and all other stimuli. This method reduced the classification problem to a standard binary task. In this way, each bin from every trial could be classified into the category of which its classifier produced the highest probability output. To evaluate decoding performance, we used the area under the ROC curve metric (AUC), which is insensitive to class imbalance as it is based on the true positive and false positive rates rather than the actual number of instances in each class. We macro-averaged the AUC for multiclass classification. The classifier was trained on 80% of the data using tenfold cross-validation, after which the AUC was estimated by testing on the remaining 20% of trials.

For backward masking, trials with similar latencies, regardless of whether the target stimulus was perceived, were first grouped into 5 groups because of an otherwise insufficient number of trials available for decoding per delay ([16, 32], [50, 66], [83, 100], [116, 132], [150, 166, 183]). A decoder was trained per delay group to classify stimulus categories. To evaluate whether awareness of the target stimulus influences decoding capacity, a separate decoder was trained for all perceived and nonperceived trials separately, combined across all delays. Next, perceived and nonperceived trials were further subdivided on the basis of similar latencies. A decoder was trained on each

subgroup to classify stimulus categories. To ensure an adequate number of perceived trials in the shortest delay groups, 3 (instead of 5) groups were created ([16, 32, 50, 66], [83, 100, 116, 132], [150, 166, 183]).

For the flash suppression and binocular rivalry experiments, a slightly modified approach was employed. First, the bin with the highest AUC in the face versus place classification, referred to as the "best bin", was identified. Then, classification for all bins per trial was performed by Leave-One-Out (LOO) cross-validation, with face being the positive class. In this process, the classifier computed the probability (face probability ranging from 0 to 1) for each bin within a specific trial—referred to as test bins—indicating how likely the bin was to belong to the face category, after being trained on the best bin across all other trials. Because this procedure trains the classifier on a specific bin across trials, but tests across all bins within a single trial, the decoder's baseline face probability is not standardized across arrays and does not necessarily center around 0.50, as would be expected in AUC-based decoding. To compare decoding results across different arrays, the baseline face probability was subtracted prior to aggregating decoding results.

For flash suppression, the stimulus presented during phase 1 was always non-rivalrous, allowing all trials to be used for decoding, as a true class could be assigned to each trial. In binocular rivalry, however, no true class could be attributed to the rivalrous conditions. Therefore, the classifier was trained exclusively on the non-rivalrous conditions. To enhance classifier performance, all stimulus alterations from the alternation paradigm were incorporated into the training process, effectively increasing the training set size by approximately 12 times for array 1 and 5 times for arrays 2 to 5. To calculate the class probability per bin for the rivalrous trials, 10 of the trained decoders that were trained on the non-rivalrous trials using LOO cross-validation, were randomly selected. Then, the class probability for all bins of all rivalrous trials was decoded by all 10 decoders, and the average over these 10 decoders was taken. To compute the probability density curve for rivalrous conditions, we first determined the probabilities for the face (1) and place class (0) for all nonoverlapping 150 ms bins across all rivalrous trials. The median probability over all rivalrous trials was subtracted for each array separately before the arrays were grouped together for calculating the probability density.

All classifications were repeated 10 times, and the average AUC or class probability was taken over these repetitions. The training and test sets were not normalized independently (cfr. "Methods"—data normalization). To compare the decoding AUC to chance performance, we computed the null AUC distribution for each time window by shuffling the trial indices 250 times. The actual decoding performance was subsequently compared to the null distribution via a one-sided permutation test ($N_{permutations} = 10000$). To compare the predicted class probabilities between groups, the analysis was repeated using 150 ms bins with a 10 ms sliding window. The class probabilities per trial within the 150 ms bin that spanned the response window were then compared between groups via a one-sided permutation test.

### Reporting summary
Further information on research design is available in the Nature Portfolio Reporting Summary linked to this article.

## Data availability
The raw data have been deposited in the figshare repository[82] at https://doi.org/10.6084/m9.figshare.28319411. The processed data are available at Code Ocean[83–87]. The underlying data for all figures are provided as a Source Data file. Source data are provided with this paper.

## Code availability
All analyses for the individual experiments (LO localizer stimuli: scrambled vs. non-scrambled, Category Selectivity, Backward

Masking, Flash Suppression, and Binocular Rivalry) are available on Code Ocean, organized as separate capsules for each experiment at https://doi.org/10.24433/CO.0628053.v1, https://doi.org/10.24433/CO.2303025.v2, https://doi.org/10.24433/CO.7059539.v1, https://doi.org/10.24433/CO.4388145.v1 and https://doi.org/10.24433/CO.5851818.v1[83–87]. Each analysis can be fully reproduced within the Code Ocean environment with a single click, automatically generating the results, datasets and figures presented in this work.

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

## Acknowledgements

This work was supported by Fonds Wetenschappelijk Onderzoek (FWO) grant G0B6422N and KU Leuven grant C14/22/134. T.T. is supported by FWO (senior clinical investigator; FWO 1830722N). M.V. holds an FWO fellowship for fundamental research (1169321N). We are grateful to all patients for their participation. We thank Stijn Verstraeten and Wouter Depuydt for technical assistance with the recording setup, Anaïs Van Hoylandt for assisting with data recording and data monitoring, Jesus Ramirez for laying the foundation for the decoders, Thomas Decramer for fine-tuning the implantation and recording technique from Utaharrays in humans and Brecht Decraene for proofreading the manuscript and providing valuable feedback.

## Author contributions

M.V. conceived and designed the experiments. T.T. and M.V. planned and performed array implantations. M.V. performed the recordings. M.V. performed all clinical trial-related activities. M.V. performed the data analysis and wrote the manuscript. T.T. and P.J. supervised and guided the study. All authors reviewed and edited the manuscript.

## Competing interests

The authors declares no competing interests.
