## [Transparent Peer Review file · Nature Communications]

Single-neuron correlates of visual consciousness in human lateral occipital complex

Corresponding Author: Dr Michaël Vanhoyland

Version 0:

Reviewer comments:

Reviewer #1

(Remarks to the Author)

This article presents results from electrophysiological single- and multiunit recordings from human lateral occipital cortex to investigate the encoding of images presented to patients in the absence of awareness. The researchers used three paradigms to render stimuli subconscious: 1. backward masking, 2. dichoptic flash suppression, and 3. binocular rivalry. The results show that neuronal activity evoked by non-perceived stimuli is much reduced and that decoding the stimulus category is significantly worse than for perceived stimuli. The general finding is probably not particularly novel considering what is known from neuroimaging. However, the temporal resolution of electrophysiology provides some insights into the dynamics that slow, indirect methods like fMRI or fNIRS could not permit, while the spatial resolution of methods like MEG or EEG is insufficient for studying encoding as precisely as this. Single-cell data in humans are difficult to come by; recordings in sites like LO is practically unheard of. As such, I feel this work should be published, given some revisions are made.

ISSUES WITH CLARITY

My main concern is that the paper is very dense and the results are hard to follow. This is particularly surprising because the general results shouldn't be that difficult to present. Part of the reason is the fact that the figures are very difficult to discern. Even though figures were not embedded in the manuscript (which as reviewer I much prefer) but only presented at the end, the font sizes and images are tiny. I had to zoom in a lot to make sense of these plots but even then it was confusing, and the clarity of the zoomed image was not great. I assume that these figures are not suitable for publication in the copy-edited version.

Some of the experiments are designed with complex paradigms (for example there are six paradigms in the flash suppression experiment). The results refer to these by number which may work well for the authors but it is difficult to follow as reviewer, and will be even more confusing to a casual reader. I suggest reworking the results presentation of all the experiments to better guide the reader through the findings and what they are supposedly showing. By the same token, what is the purpose of those radial plots in Figure 6E for the binocular rivalry data? These are certainly pretty, but it wasn't clear to me why these data are plotted in this way. What is this communicating that a linear graph or a line plot of the averages isn't?

Another related issue is that the graphs in different figures use the same colour scheme (green, red, plus sometimes blue) but to indicate different conditions. So for example, in Fig. 1B these colours refer to stimulus categories, in Fig. 3B they denote perceived vs non-perceived stimuli, in Fig. 4B they again indicate categories but in the same figure in panels C and D they denote experimental conditions - but not in the same way as in Fig. 3B. I may be misreading this (in fact, that's very likely given these issues) but my interpretation is that while 3B shows green and red as the perceived and non-perceived conditions, respectively, 4C/D does not match this arrangement conceptually? The caption states that green denotes paradigm 3 ('continued face perception') but that paradigm in Fig. 2C implies this was actually the continuous place stimulus without any rivalry in the non-dominant eye. I also note, that the colour assignment appears to be different from what is shown in Fig 4A.

The issues with Figure 4 illustrate how these numbered paradigms confuse more than they help. Wouldn't it be a better to name these conditions and explicitly point out to readers which one tests for unconscious processing?

STATISTICAL APPROACH

Similar issues with clarity are also evident for how the results are reported in the text. For starters, p-values should be reported precisely, not only relative to threshold. Moreover, for the parametric tests these should include t-values and degrees of freedom. The authors list numerous 'unpaired t-tests' for various comparisons. More commonly this is called an 'independent samples t-test' although this is perhaps a matter of taste. But it is not at all clear what the t-test actually compares. What are the observations? Patients? Trials? Neurons? (I assume not the latter but how would can we know?)

From the description in the methods it also isn't clear to me why the authors didn't use a paired t-test here. If the tests are between baseline and response windows, are these not paired measurements? Although in other places it is stated the responses were baseline corrected, in which case this is presumably a one-sample t-test against zero. But without more information, this is difficult to ascertain, let alone understand.

INTERPRETATION

The authors' main message is that 'conscious perception was required for reliable stimulus decoding'. However, as far as I could discern they did not actually test this. They show that decoding is significantly worse for unconscious than conscious stimuli but that does not mean it didn't allow reliable decoding. To test this requires a test against chance decoding. I apologise if this is included somewhere. The reporting of the results makes this very hard to work out.

Moreover, absence of evidence is not evidence of absence. The weaker response to unconscious stimuli (which is a well-known phenomenon in such studies) could simply make decoding poorer. It is also possible that fewer neurons in LO encode the non-perceived stimuli. While the decoder used should make an attempt at finding the units conveying any information, not finding accurate decoding does not rule out the possibility that only a handful of units convey meaningful information even in the absence of awareness. I don't think the authors conducted any other analyses to test this? In the same vein, how would a decoder trained on the univariate response average perform here? (Again, my apologies if these findings are included and were simply lost in the presentation).

THEORIES OF CONSCIOUSNESS

In the discussion the authors discuss their findings in the context of two popular theories of consciousness, the global neural workspace and the integrated information theory. These are not the only theories of consciousness. Considering that the authors argue that the results do not align particularly clearly with either theory, perhaps another theory is a better match altogether.

MINOR ISSUES

Figure 4 caption: Panel A is mislabelled as G. Also I find the lower case subpanels in panel A confusing although perhaps that's just me.

Figure 5: Related to the clarity issues, this figure could use a legend explaining what green and red curves refer to.

Bootstrapping: Needs more explanation. It implies data were resampled with replacement but what observations? Patients? Neurons? Trials?

Excessive use of acronyms: One example here is non-human primates, which is abbreviated as NHP, but then not used very often. As there are a lot of acronyms in this paper as it is, I would suggest getting rid of some that are not strictly needed. Again, this isn't helping clarity.

(Remarks on code availability)

I didn't see any code. The Data Availability statement mentions 'source data' being included but it's not clear what this refers to.

Reviewer #2

(Remarks to the Author)

Vanhoyland et al presented a fairly detailed description of precious single-neuron recordings in the human lateral occipital cortex.

As long as all de-identified and shareable aspects of the data are publicly and freely made available, I strongly recommend publishing this paper, although I do have some reservations about the paper itself. I believe data sharing is crucial here, partly because some of the issues in the paper are tied to the claims made in the Discussion. I understand that it might be impractical to ask the authors to conduct the relevant analyses, especially those testing integrated information theory (IIT). However, if the authors make the data accessible, someone else could perform these analyses in the future. Beyond IIT, these valuable human single-neuron recordings should be utilized to their fullest extent by the scientific community. Please see below for a series of detailed comments. My Major comments are not requirements for publication, but should be taken as suggestions for improving both the presentation of the Results and the interpretation in the Discussion.

That said, if for any reason the authors cannot make the data publicly available, I regret to say that this paper is largely descriptive and does not contribute much to understanding the mechanisms of consciousness. Under those circumstances, my Major comments would nearly become publication requirements, as they would need to be adequately addressed before the paper is accepted.

Major comments

Lack of insights into the mechanisms of consciousness, at present.

Although the authors state in Line 302,

“Our findings provide critical insights into the mechanisms underlying conscious perception and the role of LO within a broader network of brain regions.”, this conclusion is not very convincing.

Line 316: “Current evidence from human intracranial recordings predominantly supports the GNWT 9 29 34 35”

This statement is misleading. The cited studies tested the idea of GNWT, but did not explicitly contrast its predictions with those of other theories. In the context of this writing, it almost reads as though there was a direct comparison with IIT, but none of the cited works explicitly made such a comparison (see below).

“Multiple studies across different sensory modalities have reported the need for activations outside of primary sensory cortex for conscious perception 36-39”

This is misleading, as these papers mainly focused on the neural correlates of “the reports” of conscious perception. In that sense, this study truly has the potential to contribute to our understanding of “conscious perception per se” rather than just perception reports (e.g., see Aru et al 2012 Neuro Beh Res, Tsuchiya et al 2015 TICS, Block 2019 TICS). The Discussion on the ‘no-report’ condition achieved by flash suppression versus the report condition in the binocular rivalry switching is interesting, but is only briefly mentioned. Expanding that discussion would be beneficial.

Line 341: “Our results do not align entirely with the IIT, as we did not observe the sustained activity for perceived stimuli, as postulated by the IIT”

This phrasing is misleading because sustained “activity” itself is not an explicit prediction of IIT.

As discussed in REF 33 (Tononi et al 2016 NRN):

“IIT predicts that the cerebral cortex as a whole may support experience even if it is almost silent, a state which may perhaps be reached through meditative practices designed to achieve ‘naked awareness’ without content”

Unless the authors calculate something like a cause-effect structure from the neural activity patterns (based on their inferred connectivity), they have not really tested a relevant prediction of IIT.

In that regard, the authors might look at studies like Haun et al 2017 eNeuro, “Conscious Perception as Integrated Information Patterns in Human Electroencephalography,” which explicitly computed causal structures using intracranial recordings. It would be extremely interesting to test, as done by Haun et al, whether the constructed structures from part of the electrode arrays “generalize” across tasks (see their Figures 3 and 8).

In contrast to Haun et al 2017 eNeuro (also Baroni et al 2017 Neuroimage), this paper does not integrate insights from simultaneously investigating backward masking, flash suppression, and binocular rivalry. It would be best to perform such an analysis OR if a paragraph in Discussion is dedicated for that purpose.

Regarding the contextual effects on the NCC, Maier et al 2009 PNAS is quite relevant, but this paper does not incorporate it. Maier et al demonstrated that any single neuron in MT can be regarded as NCC, if paired with multiple stimuli. Although many stimuli appear to be used as suppressing or to-be-suppressed stimuli, none of these instances are analyzed.

Critical reflection on what decoding means for consciousness

It is never explicitly stated in the text what the decoding analysis reveals about consciousness. Could the authors make their assumptions more explicit? Are they implying that a linear decoding accuracy of 100% matching reported consciousness is the perfect metric for the NCC? They might consider referencing Haynes 2009 TICS or any other contemporary discussions on this topic. Discussion on the no-report paradigms in that regard is quite important.

Presentation of Results

In many figures, axis labels are too small to read. Several terms are not adequately explained (or at least not linked to the place where they are clarified). Below are some problems I found while reading:

Fig 5B: “decoding probability” is not explained. Why is Fig 5A color-coded as “prediction” (should it not say “prediction of face”)? Because it is logistic regression with leave-one-out? If “prediction of face” and “prediction of place” do not add up to 1, what does that imply?

Relatedly, Sup Fig S21: The baseline decoding probability should be explicitly indicated, such as with a horizontal dotted line showing the number of categories being predicted. As for A, Array 1, why is the face decoding probability very low at baseline? Does it reflect the prior probability of the stimulus set? (This possibility was dismissed as an attentional effect elsewhere.) And why does it go up for “place nonrivalry” in array 3 after 200 ms? Is the labeling correct? What is the x-axis? Does 0 refer to the time when the rivalry stimulus was presented (stimulus onset), or does it refer to response alternation? Generally, the baseline decoding probability is never clearly described. Yet in line 863 the authors subtract the median decoder probability to compare the arrays. This further obscures what is being plotted. What does 0 represent? What does 1 represent? If there are different numbers of categories being compared, why not use bits as a unit (e.g., 1 bit for 100% accuracy with 2 alternatives, 2 bits for 100% with 4 alternatives, etc.), or normalize so that 0 is chance and 1 is perfect decoding?

Figure 6E’s bottom panels are also unclear. It appears that the authors intended to illustrate stimulus onset information (not particularly interesting for a rivalry experiment, given Figure 6C already shows it) and switch-related information in perception.

Why not employ an ERPimage-like analysis (Jung et al 1999), for example as in Sassenhagen et al 2014, where trials are sorted (y-axis) by the duration of the Face or House period, and time (x-axis) is aligned with the switch from Face to House or vice versa?

As it stands, Figure 6E makes it hard to see exactly what each line signifies. The circular plot is not especially intuitive. What is each “spoke” meant to represent? What is the meaning of the phase angle? How were different lines arranged by phase angles?

Attentional effects

Line 134-135:

“Attentional effects were evaluated by training decoders on prestimulus baseline activity to predict whether a target stimulus would be perceived. Across all arrays, the AUC was 0.50, indicating no predictive value from baseline activity (Table S4).” This is not entirely clear. Why does this analysis necessarily reflect “attentional effects”? The Methods section further down

should be brought into the main text.

Line 715-722:

“As baseline (prestimulus) differences between perceived and nonperceived stimuli might indicate attentional effects, a decoder was trained on the 300 ms baseline from all perceived and nonperceived trials separately, as well as the 3 groups described above, to decode whether target stimuli from individual trials were perceived. An AUC value of 0.50 indicates that prestimulus activity is similar in both perceived and nonperceived trials, which minimizes the possibility of attentional effects. In contrast, an AUC value greater than 0.50 would suggest that the prestimulus state, at least partially, determines whether a stimulus will be perceived.”

Nonetheless, I am doubtful about the exact AUC value of 0.50. This could be an artifact of baseline z-normalization or a mix of units that individually show significance in different directions, canceling each other out.

Furthermore, the methods related to Sup Table S4 remain unclear. Baseline firing fluctuations (or systematic experimental design biases) should yield some distribution of AUC values and p-values for the baseline (-300 to 0 ms). With 250 shuffles, it seems implausible to get exactly AUC=0.5 and p=1 for all cases. Code and data sharing helps clarify how these values were calculated. It raises questions about the other time window results, too.

Instead, one could just use raw data and report how baseline AUC and p-values fluctuate.

Minor

A number of analyses and discussions seem only tangentially connected to consciousness (or at least their relevance is not clearly explained). I suggest cutting them and expanding on more interesting discussions, as mentioned above (e.g., the implications of no-report conditions, possible ways to compare theoretical predictions more rigorously, etc.).

Line 220-229: Eye dominance. Is this genuinely relevant or of interest here?

Sup Fig S23: What is the purpose of this figure?

Sup Fig S24: Without simultaneously presenting a perceptual report's time course, I do not see why this figure is meaningful. Especially here, the control conditions (1-3) are not informative.

Sup Fig S25: What is time 0? Is this stimulus onset?

Sup Fig S28: It is unclear why Silverman's test was used to check for response bimodality. Although I understand the approach methodologically, it is not clearly tied to consciousness.

The term “variation” is not defined or explained until line 507. You should reference it at its first mention (line 141). Even then, it is an unusual choice of wording. It appears to simply mean “experimental condition.” Possibly I am misunderstanding its use.

Sup Table S5.

I do not think “% signal loss” is the right term here. If “a positive percentage indicates that the neural response for perceived stimuli was greater than for unperceived stimuli within the specified 150 ms interval,” then a more neutral phrase such as “perceptual modulation (%)” (e.g., Wilke et al 2009) would be better.

Otherwise, “% signal” sounds like the authors assume stronger responses are the signals for conscious perception, but that assumption does not necessarily emerge from or relate to their findings. If the authors do believe this, it should be stated outright.

Line 284: Grammar?

“introduced, consciously perceived stimulus, whereas binocular input was identical in both flash”

→

“introduced a consciously perceived stimulus, in which the binocular input was identical to that in the flash suppression condition”

(Remarks on code availability)

Reviewer #3

(Remarks to the Author)

The authors recorded single- and multi-unit activity from the human lateral occipital (LO) complex under three paradigms manipulating visual awareness (backward masking, flash suppression, and binocular rivalry). Some units showed neural responses tightly linked to conscious percepts. Decoding analyses indicated that conscious perception was required for reliable stimulus prediction

Although mounting evidence (e.g. fMRI and ECoG) has indicated that neural activity in the human LO reflect conscious perception, the present study is meaningful in that the data of single-unit recordings were provided. I, however, have several concerns summarized below.

1. A definition is unclear for “non-perceived” trials of the backward masking experiment (red lines in Fig. 3B and 3C). Were they incorrect trials in which patients judged, for example, a face as an object? Or were they trials in which patients answered “Unperceived” to a target stimulus?
2. In decoding analysis of backward masking (Fig. 3C), non-perceived target (red lines at delays of 83-132 ms) induced substantial increase in decoding performance compared to a baseline period (-200 to 0 ms). This means that conscious perception was NOT necessary for reliable decoding, which contradicts the authors conclusion in Abstract (“In all three paradigms, conscious perception was required for reliable stimulus decoding from neuronal population responses”).
3. In the backward masking, patients performed a discrimination task (category judgment), not a detection task (a judgment on the presence/absence of a target). Furthermore, they were instructed not to guess in case of uncertainty (page 22). These points indicate that patients might have a residual visibility in the “non-perceived” trials (a feeling of “seeing something” but not strong enough to tell). In other words, the backward masking study might compare neural responses between high-visibility (“perceived”) and low-visibility (“non-perceived”) conditions, not between visible and invisible conditions. No data in

truly-unconscious (invisible) condition were provided.

4. In decoding analysis of the flash suppression experiment, class probabilities (vertical axis of Fig. 5) were generally biased into the “place” category (< 0.5). In all arrays, the probabilities ranged from 0.2 to 0.4 during a baseline period (-200 to 0ms). Why? They should be around 0.5.

5. The vertical axis of Figure 6C is also puzzling. In this figure, the authors plotted the decoder face probability after a median subtraction (page 31). A negative change (decrease) in face probability would be expected in “Place non-rivalry” (red) and “Place perceived rivalry” (purple) conditions, because patients perceived a place image in those trials. However, no change was actually observed in Figure 6C (even a slight increase was seen). Why?

Minor

1. Please provide the number of trials for each condition. For example, how many trials were tested in each delay (from 16 – 200 ms) in the backward masking experiment.

2. Numbers of single units in texts are slightly different from those in Supplementary Materials. For example, the authors reported “We recorded 62 visually responsive single-units on four arrays (A2-A5), 37 of which were face-selective and 7 place-selective (Table S7)” (page 8). The number of face-selective units is 34 in Table S7 (21+4+4+5 = 34).

(Remarks on code availability)

Reviewer #4

(Remarks to the Author)

The authors exploit rare multi-unit recordings from human lateral occipital cortex and probe neural correlates of visual consciousness. While the authors did an impressive job to obtain such recordings in 5 different patients with multiple paradigms to manipulate conscious vs unconscious visual processing, the paper suffers from major weaknesses.

1) The take-home message is unclear, and might even sometimes appear misleading in the current version, for two main reasons:

- The take-home message as described in the Abstract is that “conscious perception was required for reliable stimulus decoding” in the lateral occipital complex. This does not accurately summarize results: for instance this is not the case in A1 in the backward masking experiment.

- The title and beginning of Discussion both emphasize single-unit recordings, while the text mainly states that multi-unit activity (MUA) is analyzed. Ultimately, because the authors use classifiers, most of the results pertain to population activity (as appropriately mentioned in the Abstract).

2) Both the dataset and the results are quite heterogeneous

- As noted by the authors, electrodes are positioned in different parts of the lateral occipital complex (LOC). LOC was called a “complex” because indeed it is a compound of distinct areas with distinct functional properties. The different arrays of electrodes differ already in terms of latency and selectivity, but this is not really taken into account in the reasoning about visual consciousness

- Patient 1 has different results from the others in backward masking, did not do the flash suppression and his/her behavioral data in the binocular rivalry experiment seem to be weird. While it is to be expected that there are variations between patients, this seems to be quite an extreme case. In addition, it is sometimes difficult to follow the logic of analysis. For patient 1 in the binocular rivalry I understood that the patient was excluded (page 10 top§: “Patient 1 failed to differentiate between the two stimuli during rivalry conditions, which led to exclusion of the data because of uninterpretable results”), while later results are reported for this patient (page 10 last§: “Notably, this also applied to the stimulus presented to the right eye in array 1, where no dominant eye was identified”)

- Because there are many different paradigms, some with different variants, and different results in different patients, it is actually quite difficult to know what one should conclude. For instance, considering the results in Figure 5, Array 2 and Array 3 are very convincing and similar to each other, but are those 2 arrays similar in general? Then Array 4 is a kind of weaker / degraded version, and then with array 5 there is a switch to a different pattern obviously. Can this be traced down to anatomo-functional differences in array location & properties? If yes, this could really help to clarify the message – however when I tried such an approach I was left with the impression that it is unfortunately not the case

3) The results are not discussed in the context of the existing literature linking visual consciousness with activity in the lateral occipital complex. As a result, it is not entirely clear what is the novelty in this paper, but for the fact that MUA is analyzed rather than intracranial EEG or source-reconstructed data from non-invasive recordings; for a recent review see: Perceptual awareness negativity: a physiological correlate of sensory consciousness, Dembski, Cole et al. Trends in Cognitive Sciences, Volume 25, Issue 8, 660 – 670

(Remarks on code availability)

Version 1:

Reviewer comments:

Reviewer #1

(Remarks to the Author)

The authors have done a fine job addressing my previous comments. The presentation of results and statistics is a lot clearer now.

Line 252: 'its' seems to refer to data from patient 1. This should be 'their' or simply 'this patient's', if the authors want to avoid gendered language.

Line 892 (Fig 3 caption): Typo in 'indepent'

(Remarks on code availability)

Reviewer #3

(Remarks to the Author)

Thank you for revising. I have no further comments.

(Remarks on code availability)

Reviewer #4

(Remarks to the Author)

The authors' extensive revision has much improved the paper; all my previous concerns have been addressed.

(Remarks on code availability)

The code provided has almost no ReadMe file, nor comments. Besides, it includes code written by other authors (zapline) who are not credited as they should be in the reference list.

We would like to thank all 4 reviewers for thoroughly reading and reviewing the manuscript. We believe the revisions have greatly increased the quality of the manuscript. Because some changes might be of interest to all reviewers, we provide a comprehensive list of most changes that have been made:

1. As pointed out by reviewer 1, testing of significance compared to baseline should have been done with a dependent samples t-test instead of an independent samples t-test. Therefore, we redid the whole analysis, which resulted in minimal changes in the number of responsive units for some experiments. This did not affect the conclusions.
2. Major revisions have been made to the results section. The results have been written less dense with more background information so readers will find it easier to follow our thought processes (in response to reviewer 1).
3. Major revisions have been made to the discussion. Some major new topics that are discussed include: expanding on what is already known about LO in literature (in response to reviewer 4), new discussion of no-report paradigms (in response to reviewer 2), rewriting the GNWT vs IIT discussion (in response to reviewer 2) and weakening some claims that previously were overstating (in response to reviewer 1 and 4).
4. All figures have been reworked with increased resolution and new color schemes (in response to reviewer 1). To accomplish this, some figures have been split in separate figures. We included a new figure for array localization (in response to reviewer 4), put figure 6E into supplementary (in response to reviewer 1 and 2), made a new ERP-like image for the rivalry experiment (in response to reviewer 2) and brought a figure from supplementary to the main figures (new Figure 7A). Because of this, numbering of figures has changed. We therefore provide a table that links the old and new figure numbers.
5. During the revision, a paper was published in Nature from the Cogitate Consortium about the global neuronal workspace and integrated information theories of consciousness (Cogitate, C., *et al.* Adversarial testing of global neuronal workspace and integrated information theories of consciousness. *Nature* (2025).) To be able to compare some findings to our paper, we added some LFP analysis for the flash suppression experiment.

Initial manuscript	Revised manuscript
Table S1	Table S1
Table S2	Table S2
Table S3	Table S3
Table S4	Table S5
Table S5	Table S4
Table S6	Table S6
Table S7	Table S7
Table S8	Table S8
	Figure 1A (new)
Figure 1A	Figure S1
Figure 1B-E	Figure B-E
Figure 2	Figure 2

Figure 3	Figure 3
Figure 4	Figure 4
Figure 5	Figure 5
Figure 6A	Figure 6A
	Figure 6B (new)
Figure 6B	Figure S6C
Figure 6C-D + Figure S23	Figure 7A-B
Figure 6E	Figure S24
Figure S1	Figure S3
Figure S2	Figure S2
Figure S3	Figure S4
Figure S4	Figure S5
Figure S5	Figure S6
Figure S6	Figure S7
Figure S7	Figure S12
Figure S8	Figure S11
Figure S9	Figure S13
Figure S10	Figure S14
Figure S11	Figure S8
Figure S12	Figure S9
Figure S13	Figure S10
Figure S14	Figure S15
Figure S15	Figure S16
Figure S16	Figure S17
Figure S17	Figure S18
Figure S18	Figure S19
Figure S19	Figure S20
Figure S20	Figure S21
	Figure S22 (new)
Figure S21	Figure S23
Figure S22	Figure 7A
Figure S23	Figure S25
Figure S24 (removed)	
Figure S25	Figure S26
Figure S26	Figure S27
Figure S27	Figure S28
Figure S28	Figure S29

Reviewer #1 (Remarks to the Author):

This article presents results from electrophysiological single- and multiunit recordings from human lateral occipital cortex to investigate the encoding of images presented to patients in the absence of awareness. The researchers used three paradigms to render

stimuli subconscious: 1. backward masking, 2. dichoptic flash suppression, and 3. binocular rivalry. The results show that neuronal activity evoked by non-perceived stimuli is much reduced and that decoding the stimulus category is significantly worse than for perceived stimuli. The general finding is probably not particularly novel considering what is known from neuroimaging. However, the temporal resolution of electrophysiology provides some insights into the dynamics that slow, indirect methods like fMRI or fNIRS could not permit, while the spatial resolution of methods like MEG or EEG is insufficient for studying encoding as precisely as this. Single-cell data in humans are difficult to come by; recordings in sites like LO is practically unheard of. As such, I feel this work should be published, given some revisions are made.

ISSUES

WITH

CLARITY

My main concern is that the paper is very dense and the results are hard to follow. This is particularly surprising because the general results shouldn't be that difficult to present. Part of the reason is the fact that the figures are very difficult to discern. Even though figures were not embedded in the manuscript (which as reviewer I much prefer) but only presented at the end, the font sizes and images are tiny. I had to zoom in a lot to make sense of these plots but even then it was confusing, and the clarity of the zoomed image was not great. I assume that these figures are not suitable for publication in the copy-edited version.

>> We rewrote the whole results section so that it is easier to follow now. We plotted all figures again (resolution 300 dpi) and reformatted the figure so that they are suitable for publication.

Some of the experiments are designed with complex paradigms (for example there are six paradigms in the flash suppression experiment). The results refer to these by number which may work well for the authors but it is difficult to follow as reviewer, and will be even more confusing to a casual reader. I suggest reworking the results presentation of all the experiments to better guide the reader through the findings and what they are supposedly showing.

>> We changed the annotations in the plots to make the experiments easier to comprehend. We removed the numbers that referred to separate conditions.

By the same token, what is the purpose of those radial plots in Figure 6E for the binocular rivalry data? These are certainly pretty, but it wasn't clear to me why these data are plotted in this way. What is this communicating that a linear graph or a line plot of the averages isn't?

>> We plotted it this way because this is the most compact way to make a visual representation of the decoder output from all individual trials (mainly for figure 6E below).

A line plot of the average was already present in Figure 6C. Since reviewer 2 had the same remark, this figure has been brought to supplementary.

Another related issue is that the graphs in different figures use the same colour scheme (green, red, plus sometimes blue) but to indicate different conditions. So for example, in Fig. 1B these colours refer to stimulus categories, in Fig. 3B they denote perceived vs non-perceived stimuli, in Fig. 4B they again indicate categories but in the same figure in panels C and D they denote experimental conditions - but not in the same way as in Fig. 3B. I may be misreading this (in fact, that's very likely given these issues) but

my interpretation is that while 3B shows green and red as the perceived and non-perceived conditions, respectively, 4C/D does not match this arrangement conceptually? The caption states that green denotes paradigm 3 ('continued face perception') but that paradigm in Fig. 2C implies this was actually the continuous place stimulus without any rivalry in the non-dominant eye. I also note, that the colour assignment appears to be different from what is shown in Fig 4A.

>> We changed the color scheme for all figures/plots and tried to stick to the following conventions:

- Non- scrambled (dark blue) and scrambled (crimson red)
- Perceived (light green) and unperceived (light red)
- Face non-dichotopic (dark green-blue), Place non-dichotopic (yellow), face perceived dichotopic (light blue), place perceived dichotopic (light red)

We now believe the plots are easier to comprehend and follow.

The issues with Figure 4 illustrate how these numbered paradigms confuse more than they help. Wouldn't it be a better to name these conditions and explicitly point out to readers which one tests for unconscious processing?

>> We removed all numbers and annotated the figures with names of the different conditions to better show which plots represent flash suppression paradigms.

STATISTICAL

APPROACH

Similar issues with clarity are also evident for how the results are reported in the text. For starters, p-values should be reported precisely, not only relative to threshold. Moreover, for the parametric tests these should include t-values and degrees of freedom. The authors list numerous 'unpaired t-tests' for various comparisons. More commonly this is called an 'independent samples t-test' although this is perhaps a matter of taste. But it is not at all clear what the t-test actually compares. What are the observations? Patients? Trials? Neurons? (I assume not the latter but how would can we know?)

>> For all experiments, we added the degrees of freedom for the statistical tests (one-sided or two-sided permutation test; dependent or independent samples t-test). Where possible, we added the exact p-values (which isn't always possible for permutation tests when p-value is too small). We tried to better explain what is exactly compared to each other by a certain test). For example a sentence from the backward masking analysis:

"When considering only selective units, responses were 59% greater in A2 for categorical stimuli ($p = 2.71 * 10^{-11}$), 39% greater in A2 with scrambled and non-scrambled images ($p = 1.24 * 10^{-32}$), and 42% greater in A4 ($p = 1.15 * 10^{-8}$; independent samples t-test comparing net spike rate to the preferred class within a 150 ms window following response latency)"

From the description in the methods it also isn't clear to me why the authors didn't use a paired t-test here. If the tests are between baseline and response windows, are these not paired measurements? Although in other places it is stated the responses were baseline corrected, in which case this is presumably a one-sample t-test against zero. But without more information, this is difficult to ascertain, let alone understand.

>> The t-tests are indeed done on the baseline subtracted traces, which means comparing the response windows to a baseline, with mean equals to 0. It is a valid remark that baseline and response window are not independent, therefore we reran all analysis with a dependent samples t-test and changed it in the methods section. This did not alter the results. We changed this in the manuscript.

INTERPRETATION

The authors' main message is that 'conscious perception was required for reliable stimulus decoding'. However, as far as I could discern they did not actually test this. They show that decoding is significantly worse for unconscious than conscious stimuli but that does not mean it didn't allow reliable decoding. To test this requires a test against chance decoding. I apologise if this is included somewhere. The reporting of the results makes this very hard to work out.

>> This is correct. For this reason, we weakened our claims in the abstract and discussion.

Abstract: "Stimulus awareness increased decoding accuracy and decoders asserted higher probabilities to the consciously perceived stimulus during periods of dichoptic stimulus presentation."

Discussion: "Decoders more accurately classified images when they were consciously perceived during backward masking, and asserted higher probabilities to the consciously perceived stimulus during periods of dichoptic stimulus presentation in flash suppression and binocular rivalry."

Moreover, absence of evidence is not evidence of absence. The weaker response to unconscious stimuli (which is a well-known phenomenon in such studies) could simply make decoding poorer. It is also possible that fewer neurons in LO encode the non-perceived stimuli. While the decoder used should make an attempt at finding the units conveying any information, not finding accurate decoding does not rule out the possibility that only a handful of units convey meaningful information even in the absence of awareness. I don't think the authors conducted any other analyses to test this? In the same vein, how would a decoder trained on the univariate response average perform here? (Again, my apologies if these findings are included and were simply lost in the presentation).

>> It is correct that absence of evidence is not evidence of absence. It is indeed possible that fewer neurons in LO encode the non-perceived stimuli, however, since we feed the decoders with all responsive units, and the input from non-contributing units is shrunk to 0 using LASSO regularization, the decoders should prioritize any possible neurons that encode non-perceived stimuli if they are present. To be able to better test if LO contains neurons that respond to unperceived stimuli, it might have been a better way to include multiple face-place pairs within the flash suppression paradigm and decode for the unperceived/ suppressed stimulus (similarly to what has been done in the following paper (Hesse, J.K. & Tsao, D.Y. A new no-report paradigm reveals that face cells encode both consciously perceived and suppressed stimuli. *Elife* **9**(2020).) We also added some more about this in the discussion:

"However, this might be an underestimation, as we only tested a single stimulus pair.⁴³ The number of perception-modulating neurons could vary depending on the specific face-place pair used. A neuron might track perceptual changes for one such pair but not for another, even if its responses to the nonrivalrous versions of both stimuli are identical.⁴³ Moreover, the use

of a single stimulus pair limited our ability to investigate if LO activity yields sufficient information to decode perceptually suppressed stimuli from each other”

THEORIES OF CONSCIOUSNESS

In the discussion the authors discuss their findings in the context of two popular theories of consciousness, the global neural workspace and the integrated information theory. These are not the only theories of consciousness. Considering that the authors argue that the results do not align particularly clearly with either theory, perhaps another theory is a better match altogether.

>> Our electrode arrays were best positioned to evaluate IIT, as they were located in the so-called "posterior hot zone" where IIT predicts the highest level of information integration occurs. However, as noted by another reviewer, our analyses did not directly test IIT predictions, as we did not measure cause-effect structures from neural activity patterns, and 'sustained activity' by itself is not a specific prediction of IIT. For this reason, we removed the claim that our results contradict IIT.

Theories such as Higher-Order Theories (HOT) and the Global Neuronal Workspace Theory (GNWT) emphasize top-down signaling. Because we did not simultaneously record from the prefrontal cortex (PFC), we cannot make strong claims for or against these models. Nevertheless, our findings may be consistent with top-down modulation with a progressive signal loss across successive visual areas during feedforward processing, potentially reaching awareness only if a certain threshold is met in PFC. That said, HOT does not offer a better match based on our findings than GNWT or IIT.

Similarly, testing re-entry theory would require simultaneous recordings across multiple areas. Thus, we cannot conclusively evaluate this hypothesis either.

As noted in the Discussion section, during the first 1000 ms of the flash suppression experiment, we observed that initial neural responses diminished over time, even though the stimulus remained present (evidenced by reduced spike rates and decreased high-gamma activity). Despite this, stimulus identity remained decodable from LFP and SUA, even when only ~20% of neurons retained their discriminative responses. This pattern may support the structuralist theory (see Malach, *Local neuronal relational structures underlying the contents of human conscious experience*, 2021, *Neuroscience of Consciousness*), which posits that the content of conscious experience is encoded in the structure of population responses rather than their absolute magnitude.

MINOR

ISSUES

Figure 4 caption: Panel A is mislabelled as G. Also I find the lower case subpanels in panel A confusing although perhaps that's just me.

>> Changed in the manuscript. Lower case subpanels are removed. The figure has been reworked.

Figure 5: Related to the clarity issues, this figure could use a legend explaining what green and red curves refer to.

>> We changed the colors and added a legend.

Bootstrapping: Needs more explanation. It implies data were resampled with replacement but what observations? Patients? Neurons? Trials?

>> Bootstrapping was done with replacement and was done on the timestamps of the spikes corresponding to the identified peaks. We clarified this in the manuscript as follows:

“We then bootstrapped all timepoints corresponding to the identified peaks 1000 times with replacement and estimated the kernel density (KDE) of each bootstrap sample (bandwidth = 2).”

All timestamps/peaks were obtained from averaged traces that were obtained from a pool of responses from *all* (selective or responsive) *neurons to all trials corresponding to a certain delay*. Also this is further clarified in the Methods section.

Excessive use of acronyms: One example here is non-human primates, which is abbreviated as NHP, but then not used very often. As there are a lot of acronyms in this paper as it is, I would suggest getting rid of some that are not strictly needed. Again, this isn't helping clarity.

>> Changed in the manuscript (removed NHP, ASM, PEZ, PPA and FFA). Kept LO, PFC (used 8 times), ITC (used 10 times) and MTL (used 9 times).

Reviewer #1 (Remarks on code availability):

I didn't see any code. The Data Availability statement mentions 'source data' being included but it's not clear what this refers to.

>> All raw data (source data) are uploaded to figshare through the Nature Communications platform, and the analysis pipelines are uploaded to Code Ocean (where the generated datasets are available, however the analysis on Code Ocean has been done with less permutations than described in the paper/ done offline, because of limited computation time on Code Ocean).

In case the files are not available on the figshare depository of Nature Communications before publication, we also provide direct links to a personal figshare to access the data:

https://figshare.com/articles/dataset/Backward_masking_preprocessed/28764926

https://figshare.com/articles/dataset/Backward_masking/28656203

<https://figshare.com/articles/dataset/Category/28656122>

https://figshare.com/articles/dataset/Flash_Suppression/28656089

<https://figshare.com/articles/dataset/Rivalry/28656047>

https://figshare.com/articles/dataset/NSC_SCR/28645595

Reviewer #2 (Remarks to the Author):

Vanhoyland et al presented a fairly detailed description of precious single-neuron recordings in the human lateral occipital cortex.

As long as all de-identified and shareable aspects of the data are publicly and freely made available, I strongly recommend publishing this paper, although I do have some reservations about the paper itself. I believe data sharing is crucial here, partly because some of the issues in the paper are tied to the claims made in the Discussion. I understand that it might be impractical to ask the authors to conduct the relevant analyses, especially those testing integrated information theory (IIT). However, if the authors make the data accessible, someone else could perform these analyses in the future. Beyond IIT, these valuable human single-neuron recordings should be utilized to their fullest extent by the scientific community. Please see below for a series of detailed comments. My Major comments are not requirements for publication, but should be taken as suggestions for improving both the presentation of the Results and the interpretation in the Discussion. That said, if for any reason the authors cannot make the data publicly available, I regret to say that this paper is largely descriptive and does not contribute much to understanding the mechanisms of consciousness. Under those circumstances, my Major comments would nearly become publication requirements, as they would need to be adequately addressed before the paper is accepted.

Major comments

Lack of insights into the mechanisms of consciousness, at present. Although the authors state in Line 302, “Our findings provide critical insights into the mechanisms underlying conscious perception and the role of LO within a broader network of brain regions.”, this conclusion is not very convincing. Line 316: “Current evidence from human intracranial recordings predominantly supports the GNWT 9 29 34 35” This statement is misleading. The cited studies tested the idea of GNWT, but did not explicitly contrast its predictions with those of other theories. In the context of this writing, it almost reads as though there was a direct comparison with IIT, but none of the cited works explicitly made such a comparison (see below).

>> We removed both sentences and do not make any claims anymore which of either theory is best supported by current evidence. We incorporated a paper in the discussion that was published during the review process of this paper and directly compares IIT and GNWT (Cogitate, C., et al. Adversarial testing of global neuronal workspace and integrated information theories of consciousness. *Nature* (2025)).

“Multiple studies across different sensory modalities have reported the need for activations outside of primary sensory cortex for conscious perception 36-39” This is misleading, as these papers mainly focused on the neural correlates of “the reports” of conscious perception. In that sense, this study truly has the potential to

contribute to our understanding of “conscious perception per se” rather than just perception reports (e.g., see Aru et al 2012 Neuro Beh Res, Tsuchiya et al 2015 TICS, Block 2019 TICS).

>> This statement is also removed from the discussion, since it mainly made a point about the GNWT, whereas our data could be of most interest for IIT. Note however that Kronemer et al (*Kronemer, S.I., et al. Human visual consciousness involves large scale cortical and subcortical networks independent of task report and eye movement activity. Nat Commun 13, 7342 (2022)*) explicitly states that the activations are independent of task report.

The Discussion on the ‘no-report’ condition achieved by flash suppression versus the report condition in the binocular rivalry switching is interesting, but is only briefly mentioned. Expanding that discussion would be beneficial.

>> We are aware of the existence of no-report paradigms, and chose to go for a report paradigm since literature suggests that frontal, but not occipital areas are affected due to the report. We elaborated on the no-report paradigm in discussion as follows:

“A no-report paradigm, where patients are not required to report their perceptual experiences, might have been more suitable for these patients. Our flash suppression paradigm was conducted without perceptual reporting, but no-report paradigms also exist for backward masking⁵⁵ and binocular rivalry, often relying on physiological measures of pupil size, optokinetic nystagmus or position of fixation.^{11,56,57} No-report paradigms might still be susceptible to attentional biases,⁵⁸ which we attempted to mitigate by training decoders on the baseline interval, however their main benefit resides in avoiding confounds introduced by reporting-related activity. Report-related activity may mainly affect frontal rather than occipital regions.^{55,57} Therefore, while we believe a no-report paradigm could have avoided potential reporting effects, we believe report-related activity had minimal influence on our recordings.”

Line 341: “Our results do not align entirely with the IIT, as we did not observe the sustained activity for perceived stimuli, as postulated by the IIT”

This phrasing is misleading because sustained “activity” itself is not an explicit prediction of IIT.

As discussed in REF 33 (Tononi et al 2016 NRN): “IIT predicts that the cerebral cortex as a whole may support experience even if it is almost silent, a state which may perhaps be reached through meditative practices designed to achieve ‘naked awareness’ without content” Unless the authors calculate something like a cause-effect structure from the neural activity patterns (based on their inferred connectivity), they have not really tested a relevant prediction of IIT.

In that regard, the authors might look at studies like Haun et al 2017 eNeuro, “Conscious Perception as Integrated Information Patterns in Human Electroencephalography,” which explicitly computed causal structures using intracranial recordings. It would be extremely interesting to test, as done by Haun et al, whether the constructed structures from part of the electrode arrays “generalize” across tasks

(see their Figures 3 and 8). In contrast to Haun et al 2017 eNeuro (also Baroni et al 2017 Neuroimage), this paper does not integrate insights from simultaneously investigating backward masking, flash suppression, and binocular rivalry. It would be best to perform such an analysis OR if a paragraph in Discussion is dedicated for that purpose.

>> Thank you for this suggestion. We believe that this new analysis is beyond the scope of this manuscript, keeping in mind that other reviewers already mentioned that the manuscript is too densely packed. This extra analysis would further compromise readability and seems to be better suited for a possible follow-up paper. We incorporated the concerns into the discussion as follows:

“ Since we did not simultaneously record from PFC, and our analysis did not directly measure cause-effect structures from the neural activity patterns as predicted by IIT (e.g. as done by Haun *et al.*³³), we cannot draw definitive conclusions in support of either theory.”

The full discussion between GNWT and IIT has been completely reworked.

Regarding the contextual effects on the NCC, Maier et al 2009 PNAS is quite relevant, but this paper does not incorporate it. Maier et al demonstrated that any single neuron in MT can be regarded as NCC, if paired with multiple stimuli. Although many stimuli appear to be used as suppressing or to-be-suppressed stimuli, none of these instances are analyzed.

>> We believe that something similar could have been done as described by Maier in the Flash suppression or Rivalry experiment. However, we only tested 1 stimulus pair (1 face and 1 place). Therefore, we think a similar analysis on our data is not feasible. We added the following in the discussion, since we agree that this paper is relevant for our conclusions:

“Second, 64–69% of LO neurons modulated their activity in accordance with the percept during flash suppression. However, this might be an underestimation, as we only tested a single stimulus pair.⁴³ The number of perception-modulating neurons could vary depending on the specific face-place pair used. A neuron might track perceptual changes for one such pair but not for another, even if its responses to the nonrivalrous versions of both stimuli are identical.⁴³”

Critical reflection on what decoding means for consciousness
It is never explicitly stated in the text what the decoding analysis reveals about consciousness. Could the authors make their assumptions more explicit? Are they implying that a linear decoding accuracy of 100% matching reported consciousness is the perfect metric for the NCC? They might consider referencing Haynes 2009 TICS or any other contemporary discussions on this topic.

>> Decoding is a method to correlate brain activity to the report. Even with 100% accuracy, which is unfortunately never the case, one could not even be sure that the investigated brain area (LO) is ‘the true NCC’ for perception, e.g. LO could be fed information by the true NCC elsewhere (for instance, if GNWT would be true, neural activity of multiple areas such as PFC and parietal cortex could be correlated to the percept, however only the PFC would be the ‘true NCC’ since this would be the broadcaster, and without its broadcasting, the other areas would also not correlate with the percept).

We could however state, based on the results from a linear decoder, that neural activity in LO is modulated by conscious perception. We have toned down the text, also in

response to comments from the other reviewers. We now state that we observed better decoding in trials with conscious perception than in trials without conscious perception.

We stated the following at the start of the discussion:

“Decoders more accurately classified images when they were consciously perceived during backward masking, and asserted higher probabilities to the consciously perceived stimulus during periods of dichoptic stimulus presentation in flash suppression and binocular rivalry. These results offer evidence for a strong association between visual awareness and neural activity in LO, highlighting that subjective phenomenology, rather than purely physical stimulus features, predominantly drives neural responses in this region.”

At the end of the discussion, we stated the following and references to the paper from Haynes:

“Moreover, while our decoding and single-unit analyses demonstrate that neuronal activity in LO is modulated by perceptual awareness, these findings neither establish causality nor confirm that LO is necessary for the generation of conscious perception.”

Discussion on the no-report paradigms in that regard is quite important.

>> No-report paradigms are added to the discussion

Presentation of Results

In many figures, axis labels are too small to read. Several terms are not adequately explained (or at least not linked to the place where they are clarified). Below are some problems I found while reading:

Fig 5B: “decoding probability” is not explained. Why is Fig 5A color-coded as “prediction” (should it not say “prediction of face”)? Because it is logistic regression with leave-one-out? If “prediction of face” and “prediction of place” do not add up to 1, what does that imply?

>> This is a valid remark. The decoder is trained with face being the positive (true) class.

Therefore, the decoder output produces a probability that a face was presented during a certain test bin (while being trained on monocular presentation of face and place within the bin that yielded the best discrimination between both images). As such, we did not calculate a real ‘Place probability or prediction of place’ in which we assigned place to be the true class. We changed the color-coding (Fig 5A) to ‘Face probability’ to better explain the graph. We also changed the Y-axis label of Fig 5B to Face probability.

The sum of both decoders over time (one decoder with face and the other with place as true class) should always be around 1 (+/- some minimal decoder variability). As shown below, we trained for Array 2 the decoder with place being the true class (left) and face being the true class (right).

In the following plot below, the red plot signifies the sum of **face probability** within the decoder trained with face as true class. This means the sum of

- face probability when a face was presented and
- face probability when a place was presented

This is actually the sum of red and green graph before the double dotted line in the right graph above.

The blue plot below signifies the sum of face and place probability when a face was presented, it is the sum of

- face probability when a face was presented (using face as true class)
- place probability when a face was presented (using place as true class)

This is actually the sum of the red graphs (left and right plots above) before the double dotted lines. The sum of the green lines yielded the exact same results.

Importantly, the red graph is approximately 1 at the 'best' bin on which the decoder was trained on, which is to be expected. Secondly, the blue graph is constantly 1, showing that for each condition, the combined probability of it being a place or face is exactly 1.

>> Furthermore, we made the following changes to the Methods section to better clarify the decoding technique and expected baseline values: “ For the flash suppression and binocular rivalry experiments, a slightly modified approach was employed. First, the bin with the highest AUC in the face versus place classification, referred to as the "best bin", was identified. Then, classification for all bins per trial was performed by Leave-One-Out (LOO) cross-validation **with face being the positive class**. In this process, the classifier computed the probability (**face probability** ranging from 0 to 1) for each bin within a specific trial - referred to as test bins - **indicating how likely the bin was to belong to the face category**, after being trained on the best bin across all other trials. **Because this procedure trains the classifier on a specific bin across trials, but tests across all bins within a single trial, the decoder's baseline face probability is not standardized across arrays and does not necessarily center around**

0.50, as would be expected in AUC-based decoding. This is due to potential category bias in the classifier's decision boundary. To compare decoding results across different arrays, the baseline face probability was subtracted prior to aggregating decoding results.”

Relatedly, Sup Fig S21: The baseline decoding probability should be explicitly indicated, such as with a horizontal dotted line showing the number of categories being predicted. As for A, Array 1, why is the face decoding probability very low at baseline? Does it reflect the prior probability of the stimulus set? (This possibility was dismissed as an attentional effect elsewhere.) And why does it go up for “place nonrivalry” in array 3 after 200 ms? Is the labeling correct? What is the x-axis? Does 0 refer to the time when the rivalry stimulus was presented (stimulus onset), or does it refer to response alternation?

>>

- Baseline decoder activity, before stimulus onset (Time = 0 ms), represents the face probability during the 300 ms fixation period before each trial. There are always 2 categories (face or place) predicted over time (each bin), for each condition, in each array.
- We are sorry, but it is not completely clear to us what the horizontal line would need to represent exactly. If the reviewer still finds it necessary after this reply, we are willing to add the dotted horizontal line, but an example from literature might help to show us what is expected to be implemented.
- Not only for A1, but for all arrays, face probability is low during baseline. This is a direct result of the analysis method. The face class is used as true class, and the decoder is trained on the bin where the face and place stimulus are best separable (highest AUC using logistic regression 10-fold cross validation). Then, we used this trained decoder to estimate the probability that a face was presented during each bin separately. The output measure therefore is not a real classification measure (as in measuring how a decoder holds up against chance) since we did not binarize the output to make real predictions. As a result, it is expected that face probability is low during the fixation period (and not approximately 0.5).

If this question was about the fact that baseline was different between the non-rivalrous trials (green + red), compared to the rivalrous trials (purple + blue): this is because we used all the snippets from the 1 Hz alternating condition (as described in Methods) to increase the amount of data to train the decoder on. However, for A1 (not for other arrays) we used – besides the fixation period before each trial - the 300 ms before each alternation as baseline (which actually is the last 300 ms of stimulus presentation of the other stimulus). The latter shouldn't have happened and is corrected now. As a result, baseline is now again the same for all condition, however, there is a small and sudden increase for the non-rivalrous place at $t = 0$ ms as a result of this technique (which does not impact any claims). Importantly, figure S21 (new numbering of figures) shows how the decoder performs over time compared to chance level (AUC = 0.5). During baseline, AUC equals 0.5. Figure S2 is obtained from the decoding during flash suppression (not binocular rivalry), but the used stimuli are identical in both experiments.

- Response to why the face probability for place non-rivalry increases in array 3 after 200 ms: it is very difficult to find out exactly why a decoder increases the probability for face after 200 ms. since this response is based on the responses of all responsive single units. We double-checked the analysis and found no mistakes. The data was trained on the most discriminative 100 ms bin, in this case 160 ms ([110 ms – 210 ms]) after stimulus onset, as annotated with the full vertical black line in the plot below. As visualized below, the relative face probability increase for monocular place presentation is actually very small compared to the monocular face presentation, and both stimuli remain well separable. To exemplify, if a unit has an average spike rate of 10 for face and 20 for place (place-selective unit) at the best separable bin, and afterwards this specific unit diminishes its spike rate to 10, the decoder increases its face likelihood based on this unit. We don't think the decoding results for monocular face and place perception are of great importance: more important is how the decoders gauge the face probability during rivalry compared to the 'golden standard' for each arrays, which are the predictions for monocular face and place.

-
- Yes, labeling is correct.
- X-axis represents the middle of the 100 ms bin that the decoder was tested on. E.g. 0 ms holds the decoder output (face probability) from the bin 50 ms before till 50 ms after stimulus onset.
- In both A and B, $t=0$ represents the bin 50 ms before till 50 ms after stimulus onset for all rivalrous trials (not the moment of button press). For all non-rivalrous trials, $t=0$ represents the moment of stimulus onset (baseline to first stimulus), but also all alternations from the alternation paradigm (face \rightarrow place alternation is averaged in Place Non-Rivalry; place \rightarrow face alternation is averaged in Face Non-Rivalry)

Generally, the baseline decoding probability is never clearly described. Yet in line 863 the authors subtract the median decoder probability to compare the arrays. This further obscures what is being plotted. What does 0 represent? What does 1 represent? If there are different numbers of categories being compared, why not use bits as a unit (e.g., 1 bit for 100% accuracy with 2 alternatives, 2 bits for 100% with 4 alternatives, etc.), or normalize so that 0 is chance and 1 is perfect decoding?

>> This question is partially answered above. The baseline decoding probability is the probability that a face was presented during baseline (since the decoder was trained with the face being the true class).

>> We do not want to obscure the results by subtracting the median. Importantly, the median subtraction only applies to the old figure 6D/ new figure 7B. Basically, we wanted to investigate if, over prolonged periods of rivalry, a decoder trained with face as true class, would allocate higher face probability to rivalrous trials with the face presented to the dominant eye, compared to rivalrous trials with a face presented to the non-dominant eye. These results are presented in Figure S25 (new numbering). By subtracting the median, we make sure that 50% of all rivalrous time bins are <0 and 50% are >0 . This way, it is easy to visually inspect if the decoder assigns higher probability for face to bins where the face was presented to the dominant eye (if these trials are predominantly located > 0 on the histogram).

For clarity, we calculated the histograms below with mean subtraction and without subtraction, and the p-value (independent samples t-test) was highly significant for all

- Median subtraction: $p = 3.11 * 10^{-92}$
- Mean subtraction: $p = 5.05 * 10^{-92}$
- No subtraction: $p = 3.36 * 10^{-79}$

We are willing to go with any of these 3 possibilities since we believe it's just a matter of preference for visualization.

>> Importantly, there is no such measure as accuracy used in the analysis for binocular rivalry and flash suppression (which would involve thresholding the results of the face probability and assign a true binary output, for example face probability > 0.6 will be classified as face (1) and face probability < 0.6 will be classified as place (0)). Furthermore, since there is no 'true class' during rivalry trials in which a face and place are presented to opposite eyes, there is no real benefit of thresholding and assigning true-class labels, and a 'probability type' approach seems more appropriate to us.

Figure 6E's bottom panels are also unclear. It appears that the authors intended to illustrate stimulus onset information (not particularly interesting for a rivalry experiment, given Figure 6C already shows it) and switch-related information in perception. Why not employ an ERPimage-like analysis (**Jung et al 1999**), for example as in **Sassenhagen et al 2014**, where trials are sorted (y-axis) by the duration of the Face or House period, and time (x-axis) is aligned with the switch from Face to House or vice versa?

>> We believe figure 6E is mainly of interest to visualize the results for array 2, where the decoder correctly predicted a face when a face was actually perceived during rivalry. We do agree that some of these findings could be derived from figure 6C. Figure 6E is brought to supplementary and we implemented an ERP-like image as requested (new figure 6B).

As it stands, Figure 6E makes it hard to see exactly what each line signifies. The circular plot is not especially intuitive. What is each "spoke" meant to represent? What is the meaning of the phase angle? How were different lines arranged by phase angles?

>> Thank you for this suggestion. We have moved Figure 6E to supplementary. Each spoke is meant to represent a separate trial. For the upper plots, all shifts from place to face (and face to place) are also included from the 'Alternation' condition (see the revised Figure 2). This explains that the upper plots have more 'spokes' than the lower plots. The coloring is defined by the face probability during each trial over time, and the length of the spoke is defined by the moment of perceptual report in the lower plots, and fixed to 400 ms after stimulus onset for the upper plots (because the perceptual report is non-contributing there). The phase angles are meaningless: all spokes are

linearly spaces within each quadrant (so the phase angle = $\frac{90^\circ}{\text{Number of trials}}$) and are plotted in chronological order of appearance during the experiment.

Attentional effects Line 134-135:

“Attentional effects were evaluated by training decoders on prestimulus baseline activity to predict whether a target stimulus would be perceived. Across all arrays, the AUC was 0.50, indicating no predictive value from baseline activity (Table S4).” This is not entirely clear. Why does this analysis necessarily reflect “attentional effects”?

>> Given the way the experiments were performed, there is no perfect way to exclude attentional effects. By doing this analysis, we tried to exclude major pre-stimulus differences in neural activity (which could be a result of attention) between perceived and non-perceived trials. In an extreme case of daydreaming, in which the patient would be looking at the screen without paying attention in a subset of trials (resulting in not perceiving the target stimulus), pre-stimulus neural activity might be predictive whether or not the stimulus will be perceived. A similar approach was used in “van Vugt et al. The threshold for conscious report: Signal loss and response bias in visual and frontal cortex. Science”

The Methods section further down should be brought into the main text.
Line 715-722:

“As baseline (prestimulus) differences between perceived and nonperceived stimuli might indicate attentional effects, a decoder was trained on the 300 ms baseline from all perceived and nonperceived trials separately, as well as the 3 groups described above, to decode whether target stimuli from individual trials were perceived. An AUC value of 0.50 indicates that prestimulus activity is similar in both perceived and nonperceived trials, which minimizes the possibility of attentional effects. In contrast, an AUC value greater than 0.50 would suggest that the prestimulus state, at least partially, determines whether a stimulus will be perceived.”

>> This section is brought to into the results sections.

Nonetheless, I am doubtful about the exact AUC value of 0.50. This could be an artifact of baseline z-normalization or a mix of units that individually show significance in different directions, canceling each other out. Furthermore, the methods related to Sup Table S4 remain unclear. Baseline firing fluctuations (or systematic experimental design biases) should yield some distribution of AUC values and p-values for the baseline (-300 to 0 ms). With 250 shuffles, it seems implausible to get exactly AUC=0.5 and p=1 for all cases. Code and data sharing helps clarify how these values were calculated. It raises questions about the other time window results, too. Instead, one could just use raw data and report how baseline AUC and p-values fluctuate.

>> Thank you for bringing our attention to this problem. After this comment, we checked it and it was indeed an artifact of using net spike rates (baseline subtracted) (not as a result of z-normalization because the normalization is based on the response window, not baseline). We did the baseline analysis again with raw z-normalized (non-baseline subtracted) data. We also did an extra analysis and compared the average z-normalized raw baseline spike rate between perceived and unperceived stimuli and added the data to old Table S4 / new Table S5.

Minor

A number of analyses and discussions seem only tangentially connected to consciousness (or at least their relevance is not clearly explained). I suggest cutting them and expanding on more interesting discussions, as mentioned above (e.g., the implications of no-report conditions, possible ways to compare theoretical predictions more rigorously, etc.).

>> We reworked the whole discussion and elaborated more on new topics (such as no-report paradigms, known literature about consciousness and the role of LO).

Line 220-229: Eye dominance. Is this genuinely relevant or of interest here?

>> This analysis is not connected to consciousness per se, but we believe it holds value to be able to better describe and understand the neuronal properties in LO in general.

Sup Fig S23: What is the purpose of this figure?

>> This supplementary figure shows that, for each array separately (except A4), the decoder assigned higher face probabilities during rivalrous trials when the face was presented to the dominant eye compared to when the face was presented to the non-dominant eye. Figure 6D (old) / Figure 7B (new) has grouped array 2, 3, 4 and 5 together.

Sup Fig S24: Without simultaneously presenting a perceptual report's time course, I do not see why this figure is meaningful. Especially here, the control conditions (1-3) are not informative.

>> We agree that it does not add much value. The supplementary figure has been removed during the revision.

Sup Fig S25: What is time 0? Is this stimulus onset?

>> Indeed. We added it to the subscript now.

Sup Fig S28: It is unclear why Silverman's test was used to check for response bimodality. Although I understand the approach methodologically, it is not clearly tied to consciousness.

>> We agree that the investigation for a bimodal response is not linked to consciousness. The reason we checked for this is, because, to our knowledge, the only intracranial electrophysiological data on backward masking in LO (Fisch et al 2009) seems to imply that there only is a response to the target images on the population level (on LFPs). Our single-unit and multi-unit data shows that a subset of neurons also responds to the masking Mondrians (proven by the test for response bimodality, as well as by comparing spike rates within response window vs baseline on separately presented Mondrians) highlighting the importance of single- and multi-unit recordings over LFP. This also highlights heterogeneity of responses within LOC, in which only array 1 and 2 had a bimodal response pattern.

The term "variation" is not defined or explained until line 507. You should reference it at its first mention (line 141). Even then, it is an unusual choice of wording. It appears to simply mean "experimental condition." Possibly I am misunderstanding its use.

>> It indeed represents experimental condition. We now deleted this term and always stated what set of target stimuli was used.

Sup Table S5 I do not think “% signal loss” is the right term here. If “a positive percentage indicates that the neural response for perceived stimuli was greater than for unperceived stimuli within the specified 150 ms interval,” then a more neutral phrase such as “perceptual modulation (%)” (e.g., Wilke et al 2009) would be better. Otherwise, “% signal” sounds like the authors assume stronger responses are the signals for conscious perception, but that assumption does not necessarily emerge from or relate to their findings. If the authors do believe this, it should be stated outright.

>> The signal loss metric is derived from van Vugt et al. 2018. Science. We believe that it actually is appropriate here since we wanted to investigate if perceived stimuli elicited stronger responses than unperceived stimuli. We changed the wording slightly: “A positive percentage indicates that perceived stimuli elicited a larger response than unperceived stimuli within the specified 150 ms interval (see Methods)”

Line	284:	Grammar?
“introduced, consciously perceived stimulus,		
` whereas binocular input was identical in both flash”		
→		

“introduced a consciously perceived stimulus, in which the binocular input was identical to that in the flash suppression condition”

>> This sentence compares both flash suppression paradigms (face perception to place, and place perception to face through flash suppression). Therefore we believe the wording was correct. After careful consideration, this sentence was removed during rewriting of the discussion.

Reviewer #3 (Remarks to the Author):

The authors recorded single- and multi-unit activity from the human lateral occipital (LO) complex under three paradigms manipulating visual awareness (backward masking, flash suppression, and binocular rivalry). Some units showed neural responses tightly linked to conscious percepts. Decoding analyses indicated that conscious perception was required for reliable stimulus prediction

Although mounting evidence (e.g. fMRI and ECoG) has indicated that neural activity in the human LO reflect conscious perception, the present study is meaningful in that the data of single-unit recordings were provided. I, however, have several concerns summarized below.

1. A definition is unclear for “non-perceived” trials of the backward masking experiment (red lines in Fig. 3B and 3C). Were they incorrect trials in which patients judged, for example, a face as an object? Or were they trials in which patients answered “Unperceived” to a target stimulus?

>> Unperceived meant that they assigned a wrong class to the stimulus, or that they did not respond at all. Patients were not instructed to verbally report ‘unperceived’ in

case they didn't perceive the stimulus. To clarify this, we added the following in the methods section: "A stimulus was deemed unperceived in case the target stimulus was categorized wrongly, or if the patient did not make a verbal report."

2. In decoding analysis of backward masking (Fig. 3C), non-perceived target (red lines at delays of 83-132 ms) induced substantial increase in decoding performance compared to a baseline period (-200 to 0 ms). This means that conscious perception was NOT necessary for reliable decoding, which contradicts the authors conclusion in Abstract ("In all three paradigms, conscious perception was required for reliable stimulus decoding from neuronal population responses").

>> We agree with this observation. We do not exclude that some trials that were actually perceived ended up in the non-perceived group due to the cognitive incapacities of the subjects to correctly categorize (we are investigating patients with long-lasting intractable epilepsy here). Even within the longest delays (150 – 183 ms), there were some 'unperceived' trials, which is not to be expected. Testing in our lab with healthy volunteers yielded almost 100% correctness. It is possible that the significant decoding from baseline within these groups stems from residual perception. We added more text in discussion for this and weakened our claim in the abstract.

3. In the backward masking, patients performed a discrimination task (category judgment), not a detection task (a judgment on the presence/absence of a target). Furthermore, they were instructed not to guess in case of uncertainty (page 22). These points indicate that patients might have a residual visibility in the "non-perceived" trials (a feeling of "seeing something" but not strong enough to tell). In other words, the backward masking study might compare neural responses between high-visibility ("perceived") and low-visibility ("non-perceived") conditions, not between visible and invisible conditions. No data in truly-unconscious (invisible) condition were provided.

>> Same answer as above (2).

4. In decoding analysis of the flash suppression experiment, class probabilities (vertical axis of Fig. 5) were generally biased into the "place" category (< 0.5). In all arrays, the probabilities ranged from 0.2 to 0.4 during a baseline period (-200 to 0ms). Why? They should be around 0.5.

>> The decoder is trained on the best-bin (Figure S20), and the metric here is not AUC (which indeed would need to be around 0.5). What we visualize is the raw output of the logistic regression with LOO cross validation, which estimates the probability that a certain bin belongs to the positive class (= face). Therefore, the value between 0 and 1 should be interpreted as the probability that a face was shown during a certain bin (which should be low during baseline. Since the bin is trained on the best-bin, which for most channels means a high spike rate for face and low (or even 0) net spike rate for place, it is to be expected that baseline activity resembles activity of the place image more than the face image. We added the following (in red) to the methods section for clarity:

"Then, classification for all bins per trial was performed by Leave-One-Out (LOO) cross-validation with face being the positive class. In this process, the classifier computed the probability (ranging from 0 to 1) for each bin within a specific trial - referred to as test bins - indicating whether the bin belonged to the face class (1) or the place class (0), after being trained on the best bin across all other trials.

"

5. The vertical axis of Figure 6C is also puzzling. In this figure, the authors plotted the decoder face probability after a median subtraction (page 31). A negative change (decrease) in face probability would be expected in “Place non-rivalry” (red) and “Place perceived rivalry” (purple) conditions, because patients perceived a place image in those trials. However, no change was actually observed in Figure 6C (even a slight increase was seen). Why?

>> The median subtraction applies to figure 6D, in figure 6C (old numbering), the mean baseline decoding is subtracted per array, resulting in an average baseline of 0. We added the following sentence to make this clear within the text accompanying figure 6: “Before grouping, mean baseline decoding probability was subtracted for all conditions per array.” Figure 6C (old)/7A (new) displays the average from the plots presented in figure S23 (new numbering). Mainly for array 3 and 5 (not for 2 and 4), there is some increase in face probability during rivalry even though the first perceived image was the face. This could be explained by a couple of factors. Firstly, the same reasoning as above in point 4, namely that ‘face probability’ during baseline is low by itself (as an example, if the best decodable bin has an average net spikerate of 10 for face and 2 for place, and average net spikerate during baseline is 0, the decoder will assign lower face probabilities to baseline than to place perception).

Secondly, there still could be some residual perception for the unreported face stimulus during ‘face perceived rivalry’ (for which we cannot check during this experiment, therefore we did the flash suppression experiment), even though the first perception was a place. Also note that the duration after which the patients reported their first perception varied (as seen in the old radial plot Fig 6E – bottom at is now brought to supplementary): e.g. it might be that, if a patient first reported 2000 ms after stimulus onset, there might be some “in between” perception between face and place at first, compared to if the patient reported their first perception almost immediately after 1000 ms. The main goal of this figure was to show that the decoder output seems to align with the perceived image, rather than the visual input (which is identical for the binocular rivalry trials).

Minor

1. Please provide the number of trials for each condition. For example, how many trials were tested in each delay (from 16 – 200 ms) in the backward masking experiment.

>> This has been implemented now.

2. Numbers of single units in texts are slightly different from those in Supplementary Materials. For example, the authors reported “We recorded 62 visually responsive single-units on four arrays (A2-A5), 37 of which were face-selective and 7 place-selective (Table S7)” (page 8). The number of face-selective units is 34 in Table S7 (21+4+4+5 = 34).

>> Thank you for pointing this out. The number in the main text was correct. We changed the numbers in Table S7 (24 + 4 + 4 + 5).

Reviewer #4 (Remarks to the Author):

The authors exploit rare multi-unit recordings from human lateral occipital cortex and probe neural correlates of visual consciousness. While the authors did an impressive job to obtain such recordings in 5 different patients with multiple paradigms to manipulate conscious vs unconscious visual processing, the paper suffers from major weaknesses.

1) The take-home message is unclear, and might even sometimes appear misleading in the current version, for two main reasons:

- The take-home message as described in the Abstract is that “conscious perception was required for reliable stimulus decoding” in the lateral occipital complex. This does not accurately summarize results: for instance this is not the case in A1 in the backward masking experiment.

>> We weakened our claims in the abstract and discussion to “Decoders more accurately classified images when they were consciously perceived during backward masking, and asserted higher probabilities to the consciously perceived stimulus during periods of dichoptic stimulus presentation in flash suppression and binocular rivalry. These results offer evidence for a strong association between visual awareness and neural activity in LO”.

- The title and beginning of Discussion both emphasize single-unit recordings, while the text mainly states that multi-unit activity (MUA) is analyzed. Ultimately, because the authors use classifiers, most of the results pertain to population activity (as appropriately mentioned in the Abstract).

>> We only analyzed MUA for the “Visual Selectivity” section since there was no real benefit of SUA over MUA there. All other analysis were based on SUA. We agree that using classifiers on SUA, this loses useful information that SUA hold over MUA and leads to an analysis of population responses. For that reason, we included SUA example units in flash suppression and binocular rivalry, plotted SUA examples in supplementary for backward masking and performed SUA spike rate analysis for flash suppression, including calculations for percentage of perception tracking neurons.

2) Both the dataset and the results are quite heterogenous
- As noted by the authors, electrodes are positioned in different parts of the lateral occipital complex (LOC). LOC was called a “complex” because indeed it is a compound of distinct areas with distinct functional properties. The different arrays of electrodes differ already in terms of latency and selectivity, but this is not really taken into account in the reasoning about visual consciousness

>> The dataset indeed is heterogenous in terms of functional properties of the recording sites, as well as the performed experiments across subjects. Unfortunately, this is a direct result from micro-electrode recordings in epileptic patients.

Nonetheless, we believe most of the reported results are replicated in most or all arrays. While this functional heterogeneity at first sight might seem as a drawback, it actually shows the importance of these recordings since these differences might never have been discovered with conventional techniques such as ECoG / MEG. As further explained below, we took the selectivities and latencies into account to draw conclusions. More particularly, array 1 was positioned in a completely different subregion of LO, thereby explaining the different results compared to the other 4 arrays.

- Patient 1 has different results from the others in backward masking, did not do the flash suppression and his/her behavioral data in the binocular rivalry experiment seem to be weird. While it is to be expected that there is variations between patients, this seems to be quite an extreme case. In addition, it is sometimes difficult to follow the logic of analysis. For patient 1 in the binocular rivalry I understood that the patient was excluded (page 10 top§: “Patient 1 failed to differentiate between the two stimuli during rivalry conditions, which led to exclusion of the data because of uninterpretable results”), while later results are reported for this patient (page 10 last§: “Notably, this also applied to the stimulus presented to the right eye in array 1, where no dominant eye was identified”)

>> Patient 1 was chronologically the first patient that was implanted with a Utah array and on which we tested the experiments for the first time. Its preliminary analysis and feedback lead to changes for the other patients, such as reducing stimulus duration from 24 to 10 seconds during binocular rivalry.

>> The exclusion of the data referenced to the perceptual related data (e.g. the button presses during rivalry). Other analyses were still valid and therefore reported. We changed the sentence for clarity: “Patient 1 failed to differentiate between the two stimuli during rivalrous conditions. Therefore, we do not present its data in relation to the perceptual report.”

>> We still believe that, despite its flaws in methodology (not performing flash suppression and missing perceptual data during rivalry), this patient is a valuable addition to the manuscript because it sheds lights on different response patterns within

LOC. Furthermore, we start out results section with providing evidence that its responses are electrophysiologically different from the other arrays: this array had very short latencies, its selectivity for non scrambled > scrambled is only weak and restricted to the last part of the response, while the initial response is identical between the scrambled and non-scrambled images. On a typical LO localizer in fMRI, this array would probably be within the typical LO hotspot (or on the very edge), yet we did not observe significant decoding for different categories. We added the following to the discussion in respect of patient 1: “Although we were unable to perform flash suppression in array 1, and its activity related to the actual percept was uninterpretable during binocular rivalry, it nonetheless offered valuable insights. Analysis of its response properties indicated that array 1, despite showing a stronger response to non-scrambled images, was positioned in a distinct, more inferoposterior region of LO compared to the other arrays, reflecting early visual responses (Figure 1A). This likely explains its strong responsiveness to the masking Mondrians and the inability to reliably decode the perceived stimulus category during backward masking.”

>> Below, we plotted all 5 arrays on a high-resolution MNI atlas of the brain. Array 1 is positioned very low and posterior within LOC. Note that also array 2 is located low in LO (yet a bit higher). We believe that there is a clear link between the anatomical location and observed response latencies. Those 2 arrays exhibited fast response latencies (A1: 65 ms, A2: 85 ms) and exhibited responses to the masking Mondrians during backward masking. Furthermore, array 3 and 4 finish a caudo-rostral gradient of response latencies (A3: 130 ms, A4: 200 ms) . Array 5 (140 ms) does not fit in this gradient. The 2 most rostral arrays (array 4 and 5) show categorical responses.

- Because there are many different paradigms, some with different variants, and different results in different patients, it is actually quite difficult to know what one should conclude. For instance, considering the results in Figure 5, Array 2 and Array 3 are

very convincing and similar to each other, but are those 2 arrays similar in general? Then Array 4 is a kind of weaker / degraded version, and then with array 5 there is switch to a different pattern obviously. Can this be traced down to anatomo-functional differences in array location & properties? If yes, this could really help to clarify the message – however when I tried such an approach I was left with the impression that it is unfortunately not the case

>> The previous answer also applies here. We believe that many differences between arrays are a result of different anatomical localizations. Importantly, we are working with human patients. Therefore, we cannot implant the arrays to its ideal and identical position across patients, since the location of the arrays are dependent on the clinical needs.

3) The results are not discussed in the context of the existing literature linking visual consciousness with activity in the lateral occipital complex. As a result, it is not entirely clear what is the novelty in this paper, but for the fact that MUA is analyzed rather than intracranial EEG or source-reconstructed data from non invasive recordings; for a recent review see: Perceptual awareness negativity: a physiological correlate of sensory consciousness, Dembski, Cole et al. Trends in Cognitive Sciences, Volume 25, Issue 8, 660 – 670

>> We added a section to the discussion, trying to discuss our results in relation to existing literature in regard of what is known from activity in LO.

Point by point answers to the reviewers

We would like to thank all 3 reviewers for thoroughly and quickly reviewing the manuscript after the revision. We are very pleased that all reviewers are now satisfied with the revised manuscript.

Reviewer #1 (Remarks to the Author):

The authors have done a fine job addressing my previous comments. The presentation of results and statistics is a lot clearer now.

Line 252: 'its' seems to refer to data from patient 1. This should be 'their' or simply 'this patient's', if the authors want to avoid gendered language.

- We changed this to: "this patient's data"

Line 892 (Fig 3 caption): Typo in 'independent'

- The typo has been rectified.

Reviewer #3 (Remarks to the Author):

Thank you for revising. I have no further comments.

Reviewer #4 (Remarks to the Author):

The authors' extensive revision has much improved the paper; all my previous concerns have been addressed.

Reviewer #4 (Remarks on code availability):

The code provided has almost no ReadMe file, nor comments. Besides, it includes code written by other authors (zapline) who are not credited as they should be in the reference list.

- We now included a more extensive ReadMe file for the analysis on Code Ocean, giving a better overview of the functions used in other functions, and clarifying what each function exactly calculates, including which figures are plotted.
- We are sorry about this mistake. Zapline has been credited in the reference list now.

de Cheveigne, A. ZapLine: A simple and effective method to remove power line artifacts. *Neuroimage* **207**, 116356 (2020).

<https://doi.org/10.1016/j.neuroimage.2019.116356>